# Temperature compensation in a small rhythmic circuit

Leandro M Alonso*, Eve Marder

Volen Center and Biology Department, Brandeis University, Waltham, United States

**Abstract** Temperature affects the conductances and kinetics of the ionic channels that underlie neuronal activity. Each membrane conductance has a different characteristic temperature sensitivity, which raises the question of how neurons and neuronal circuits can operate robustly over wide temperature ranges. To address this, we employed computational models of the pyloric network of crabs and lobsters. We produced multiple different models that exhibit a triphasic pyloric rhythm over a range of temperatures and explored the dynamics of their currents and how they change with temperature. Temperature can produce smooth changes in the relative contributions of the currents to neural activity so that neurons and networks undergo graceful transitions in the mechanisms that give rise to their activity patterns. Moreover, responses of the models to deletions of a current can be different at high and low temperatures, indicating that even a well-defined genetic or pharmacological manipulation may produce qualitatively distinct effects depending on the temperature.

## Introduction

Biological systems depend on many interacting nonlinear processes that together produce complex outputs. In the nervous system, neuronal activity requires the coordinated activation and inactivation of many inward and outward currents. Temperature influences all biological processes, to a greater or lesser degree. This poses an inherent difficulty for neuronal signaling: if the currents involved in neuronal and network dynamics are differentially temperature-dependent a system that is well-tuned to work at one temperature may not function at a different temperature (*Caplan et al., 2014*; *O'Leary and Marder, 2016*; *Tang et al., 2010*; *Tang et al., 2012*). Nonetheless, many ectothermic animals have neurons and circuits that function well over an extended temperature range (*Robertson and Money, 2012*). It then becomes important to understand how and to what extent this can occur.

The effects of temperature on channel function and neuronal activity have been studied extensively for many years (*Taylor and Kerkut, 1958*). Increasing temperature generally results in increases of channel maximal conductances and faster activation/inactivation rates (*Frankenhaeuser and Moore, 1963*). But, importantly ionic channels of different types are affected by temperature to different extents: each of these processes has a different $Q_{10}$ (*Schauf, 1973*; *Kukita, 1982*; *Ruff, 1999*; *Tang et al., 2010*; *Ranjan et al., 2019*). The effect of temperature on neuronal intrinsic excitability such as voltage and current thresholds can be diverse (*Sjodin and Mullins, 1958*; *Guttman, 1962*; *Guttman, 1966*; *Fitzhugh, 1966*). For example, identified neurons in locust showed reversible changes in spike amplitude and duration of spikes as temperature varies from 18 to 35°C (*Heitler et al., 1977*). Studies in the jump neural circuit of grasshopers showed that neural excitability is differently affected by temperature across neuronal types (*Abrams and Pearson, 1982*).

There are also examples of neuronal and circuit processes that are relatively temperature-compensated. The frequency/current (f/I) curves in both molluscan and locust neurons are not substantially affected by small temperature changes (6–8°C) (*Connor, 1975*; *Heitler et al., 1977*). The f/I

*For correspondence:
lalonso@brandeis.edu

Competing interests: The authors declare that no competing interests exist.

curves of auditory sensory neurons in grasshoppers remain largely unaffected by changes in temperature between 21°C and 29°C (*Roemschied et al., 2014*). Temperature compensation also takes place across the behavioral level. For example, a recent study in hunting archerfish showed that the duration of the two major phases of the C-start—a fast escape reflex which follows prey release—were temperature compensated (*Krupczynski and Schuster, 2013*). The question then arises of how these behaviors are preserved when the currents in the cells and their intrinsic excitability properties are differentially modified by temperature.

Here, we explore these issues using computational models of the pyloric network—a subnetwork within the stomatogastric ganglion (STG) of crustaceans (*Marder and Bucher, 2007*; *Maynard, 1972*). The pyloric rhythm is a triphasic motor pattern that consists of bursts of action potentials in a specific sequence. This behavior is robust and stable, and the cells and their connections are well-characterized. Robustness to temperature in the pyloric network has been explored experimentally by *Tang et al., 2010*; *Tang et al., 2012*; *Soofi et al., 2014* and *Haddad and Marder, 2018*. These studies show that as temperature increases the frequency of the pyloric rhythm increases but the phases of the cycle at which each cell is active remain approximately constant. Additional work on the pacemaker kernel of the pyloric network (three cells connected by gap junctions that burst synchronously) showed that temperature increases the frequency of these bursts ($Q_{10} \approx 2$), but their duty cycle (the burst duration in units of the period) stays approximately constant (*Rinberg et al., 2013*).

Temperature robustness was explored in computational models of the pacemaking kernel (*Soto-Treviño et al., 2005*) by *Caplan et al., 2014* and *O'Leary and Marder, 2016*. They showed that it was possible to find multiple sets of $Q_{10}$ values for different membrane conductances processes, so that the duty cycle of the cells remained constant as their bursting frequency increased. In this work, we build on the results in *Caplan et al., 2014* and implemented temperature sensitivity in a model of the pyloric network (*Prinz et al., 2004*). We show that in these models, there are multiple sets of maximal conductances and temperature sensitivities that reproduce much of the experimental phenomenology previously reported (*Tang et al., 2010*; *Tang et al., 2012*; *Soofi et al., 2014*; *Haddad and Marder, 2018*). In addition, we explored how the dynamics of the currents are modified to sustain the correct activity at each temperature. We performed this study for 36 different models and found that in all cases, the contributions of the currents to the activity can be significantly different across temperatures. The currents are not simply scaled up but instead become reorganized: a current that is important for burst termination at 10°C may no longer play that role at 25°C. Because the contribution of a given current to a neuronal process can be replaced by another at different temperatures, deletion or blockade of a current can produce qualitatively different effects at high and low temperatures. These results provide a plausible hypothesis for why interactions between temperature and a second perturbation can be observed experimentally (*Haddad and Marder, 2018*; *Ratliff et al., 2018*).

## Results

### Duty cycle and phase maintenance in model pyloric networks

The triphasic pyloric rhythm is produced by the periodic sequential activation of the pyloric dilator (PD) neurons, which are electrically coupled to the anterior burster (AB) neuron forming a pacemaking kernel, the lateral pyloric (LP) neuron and five to eight pyloric (PY) neurons. We modified the model of the pyloric circuit in *Prinz et al., 2004* to include temperature sensitivity. *Figure 1A* shows a schematic representation of the model pyloric network studied here. The model network consists of three cells, each modeled by a single compartment with eight currents as in previous studies (*Golowasch and Marder, 1992*; *Buchholtz et al., 1992*; *Goldman et al., 2001*). The synaptic connections are given by seven chemical synapses of two types as in *Prinz et al., 2004*. In this model the pacemaking kernel is aggregated into a single compartment $AB - PD$ (here we refer to this compartment as *PD*). Following *Liu et al., 1998*, each neuron has a sodium current, $I_{Na}$; transient and slow calcium currents, $I_{CaT}$ and $I_{CaS}$; a transient potassium current, $I_A$; a calcium-dependent potassium current, $I_{KCa}$; a delayed rectifier potassium current, $I_{Kd}$; a hyperpolarization-activated inward current, $I_H$; and a leak current $I_{leak}$. The traces in *Figure 1A* show a solution of this model for one set of maximal conductances **G**. The traces exhibit a triphasic pyloric rhythm that consists of the

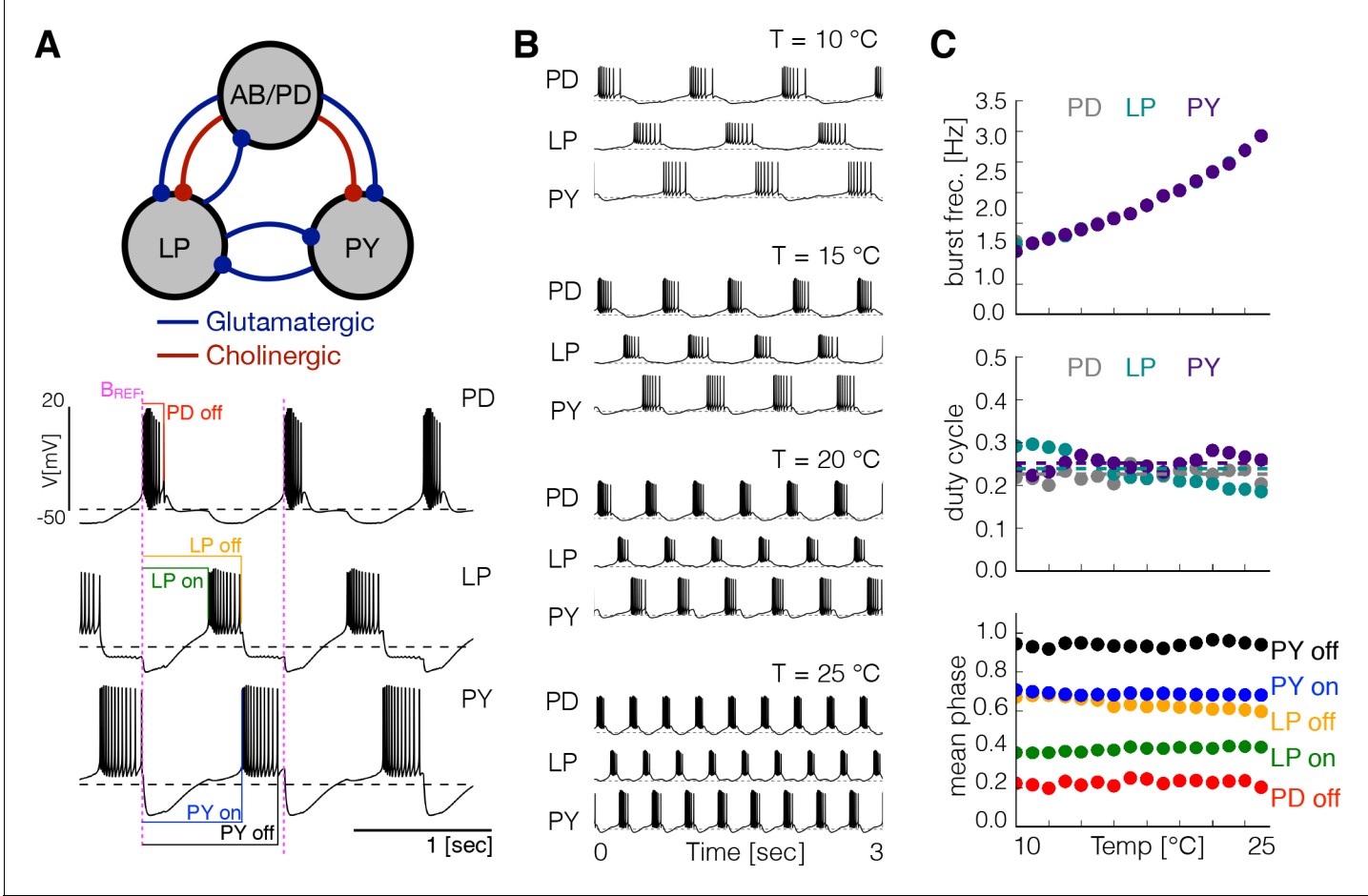

**Figure 1.** Effects of temperature in a model pyloric network. (**A**) Schematic diagram of the model pyloric network in *Prinz et al., 2004*. The three groups interact via seven inhibitory chemical synapses. The red synapses are cholinergic from the PD neurons and all others are glutamatergic. The traces below show a representative solution that exhibits a triphasic rhythm: the activity is approximately periodic and the cells burst in a specific sequence: *PD-LP-PY*. (**B**) Activity of a temperature robust model network at 10°C - 25°C. As temperature increases the frequency of the rhythm increases but the duty cycle of the cells (the burst duration in units of the period) remains approximately constant. (**C**) Top: average burst frequency of each cell over temperature (values are nearly identical so dots overlap). Middle: average duty cycle of each cell (cell type indicated in colors). Bottom: average phases of the cycle at which bursts begin and terminate using the start of the PD burst as reference (indicated by label $B_{REF}$ in magenta). As temperature increases these phases remain approximately constant. The panels show average values over 30 s for 16 values of temperature between 10°C - 25°C.

The online version of this article includes the following figure supplement(s) for figure 1:

**Figure supplement 1.** Equivalent phenotype for different sets of maximal conductances and temperature sensitivities.

**Figure supplement 2.** Spiking patterns during temperature ramps.

**Figure supplement 3.** Duty cycle distributions of all models.

sequential bursting of the *PD*, *LP* and *PY* cells. This pattern remains approximately periodic so we can measure the phases of the cycle at which each burst begins and terminates using the burst start of the *PD* cell as reference (indicated as $B_{REF}$ in magenta in *Figure 1A*), as indicated by the color labels.

Temperature sensitivity was modeled for each conductance as in *Caplan et al., 2014* and *O'Leary and Marder, 2016* (see Materials and methods). We increased each conductance by an Arrhenius type factor $R(T) = Q_{10}^{\frac{T-T_{ref}}{10}}$ (with $Q_{10} \geq 1$ and $T_{ref}$ = 10°C) and also increased the rates of the channel kinetics $\tau$ in similar fashion. We assume that $Q_{10_G} \in [1, 2]$ for the maximal conductances and $Q_{10_\tau} \in [1, 4]$ for the timescales as these ranges are consistent with experimental measurements of these quantities (*Tang et al., 2010*). The $Q_{10}$ values are a property of the channel proteins, so in a

given model we use the same $Q_{10}$ for a given current (and synapse) type in all cells. Different values of the maximal conductances $\mathbf{G}$ can produce an innumerable variety of activities. The space of solutions of the model—the different patterns produced for each set of maximal conductances $\mathbf{G}$—is formidably complex. This is in part due to the nonlinear nature of neuronal dynamics but also because the number of conductances that need to be specified is large (31 in total, 8 intrinsic conductances ×3 cells +7 synaptic conductances). The picture is more complicated as we consider the effect of temperature because it affects both the conductances and the time scales of the different processes (14 in total: 12 intrinsic, 2 synaptic). Therefore in the models, changes in temperature correspond to a coordinated change or a path in a 45-dimensional parameter space. The complexity of this problem led us to confine this study to specific questions suggested by biological observations over a permissible temperature range.

In the crab, as temperature is increased pyloric activity remains triphasic—the cells fire in the same order and at the same relative phases—while the pyloric network frequency increases by a two- to three-fold factor over a 15℃ range (*Tang et al., 2010*). For this to happen in the model, there should be many regions in the parameter space for which the activity is triphasic at different frequencies and paths in the parameter space (sequences of values of $\mathbf{G}$ and $\boldsymbol{\tau}$) that interpolate these regions. One of the main results in this work is that such paths do exist and that temperature compensation is possible in this model. Finding these temperature compensated networks is nontrivial: it requires the specification of 31 maximal conductances $\mathbf{G}$ and 24 $\mathbf{Q_{10}}$ values (see Materials and methods). Because it is virtually impossible to find these sets of parameters at random we employed a landscape optimization scheme described previously (*Alonso and Marder, 2019*) and adapted it to this particular scenario (see Materials and methods). We first searched for sets of maximal conductances $\mathbf{G}$ that displayed triphasic rhythms at control temperature (10℃). Inspired by physiological recordings of the pyloric rhythm of crabs and lobsters we targeted control activities with a network frequency of $1\,Hz$, duty cycle $\approx$ 20% for *PD*, and duty cycle $\approx$ 20% for *LP* and *PY* (*Bucher et al., 2006*; *Hamood et al., 2015*). We then selected 36 of these models and for each of them we searched for $\mathbf{Q_{10}}$ values so that the activity is preserved over a temperature range. We found that, regardless of the maximal conductances $\mathbf{G}$, it was always possible to find multiple sets of $\mathbf{Q_{10}}$ values that produced a temperature robust pyloric rhythm.

*Figure 1B* shows the activity of one example network at several temperatures, similar to the experimental results reported by *Tang et al., 2010*. *Figure 1C* shows the same analysis as in *Tang et al., 2010* performed on this particular model. We simulated the model at several temperatures over the working range (10℃ to 25℃) and computed the average duty cycle, frequency and phases of the bursts. While the burst frequency increases with $Q_{10} \approx 2$, the duty cycle of the cells, and the phases of the onsets/offsets of activity, remain approximately constant. *Figure 1—figure supplement 1A* shows the same analysis performed on three different models. In all cases, the duty cycle and the phases stay approximately constant while the frequency increases. *Figure 1—figure supplement 1B* shows the values of the maximal conductances $\mathbf{G}$ and the $\mathbf{Q_{10}}$ values for each of these models.

While the duty cycle remains approximately constant on average, the spiking patterns of the cells show discernible differences across temperatures. We subjected the models to temperature ramps from 10℃ to 25℃ over 60 min and recorded the spike times (*Figure 1—figure supplement 2*). As temperature is increased the spiking patterns change in complicated ways and the largest *ISI* decreases monotonically, consistent with the overall increase in bursting frequency of the cells. Different values of temperature result in slightly different spiking patterns which in turn result in different values of the duty cycle. For some temperatures, the duty cycle is nearly identical in every burst resulting in a single point in the y-axis (*Figure 1—figure supplement 2D*, blue box in PD cell). There are temperatures for which the duty cycle takes two values preferentially (*Figure 1—figure supplement 2D*, pink box in *PD* cell) and there are temperatures for which the duty cycle differs noticeably from burst to burst. Although in all models, the average duty cycle and phases stay approximately constant, as temperature is changed the precise way in which the spiking patterns change show marked differences across models: different maximal conductances and different $Q_{10}$ values produce different patterns. *Figure 1—figure supplement 3* shows the temperature dependence of the duty cycle in the *LP* cell over the working range for all 36 models and provides a notion of how variable the bursting patterns of the cells can be across temperatures and individuals.

## Changes in membrane potential over temperature

Experimental studies on other systems reported measurable effects of temperature on resting membrane potentials (*Klee et al., 1974*), spiking thresholds, and amplitudes of spikes (*Heitler et al., 1977*). In the models studied here, some temporal properties of the activity remain approximately constant over a temperature range but the precise shape of the waveforms can also change. To inspect how the membrane potential changes over the working temperature range (10°C - 25°C), we computed the distribution of $V$ of each cell for 101 values of temperature. Temperature changes features of the voltage waveforms, resulting in changes in these distributions. *Figure 2* shows example traces and the result of this analysis for the *LP* cell in one model. The membrane potential changes with temperature in several ways: the spike amplitude decreases and the spike threshold depolarizes as temperature increases. At each temperature, we computed the distribution of $V$, and used a gray scale to indicate the fraction of time that the cell spends at each voltage value (y-axis). The color lines match features of the voltage waveforms at each temperature, with features in the distribution on the right. The distributions permit visualizing changes in the waveform as the control parameter is changed (*Alonso and Marder, 2019*). The effect of temperature on membrane potential is not consistent across models and depends on the precise values of conductances and temperature sensitivities. *Figure 2—figure supplement 1* shows the membrane potential distributions of each cell for three example models. In all cases, the waveforms show visible differences across temperatures and there is considerable variability in the amplitudes of the oscillations across models. In some models, the peak voltage of spikes, as indicated by the upper envelope of the distributions, decreases (left), increases (middle), or remains relatively constant (right). In addition, we found that cells can either become more depolarized or more hyperpolarized as temperature increases.

## Dynamics of the currents at different temperatures

Neuronal activity is governed by currents that result from the precise activation and inactivation of ion channels with different kinetics. Experimental access to these quantities is limited because it is hard to measure currents individually without blocking all other currents, and thus changing the activity. Here, we employ models to explore how these currents are differentially altered by

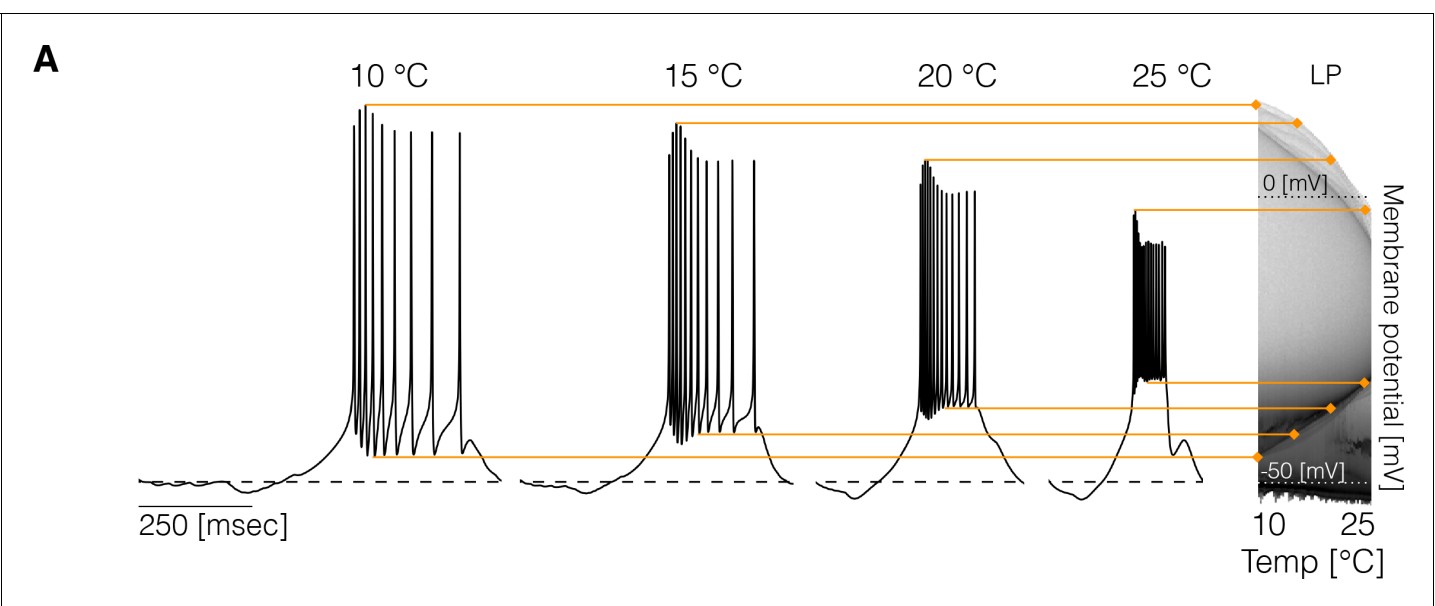

**Figure 2.** Changes in membrane potential over temperature. (left) Representative traces of the *LP* cell in one model at different temperatures. (right) Membrane potential distribution at each temperature. The gray scale indicates the proportion of time the cell spends at that potential. The distribution facilitates inspecting how features such as the total amplitude of the oscillations and spiking thresholds change with temperature.

The online version of this article includes the following figure supplement(s) for figure 2:

**Figure supplement 1.** The panels show the membrane potential distributions of each cell over temperature for three models.

temperature, and yet are able to remain balanced in such a way that the network activity speeds up while preserving its behavior and phase relationships.

We first compared the dynamics of the currents in two model *PD* neurons with different maximal conductances **G** and temperature sensitivities **Q₁₀** by direct inspection of the currents' time series. *Figure 3A* shows the time series of each intrinsic current in *PD* at 10°C and 25°C for one model. The corresponding membrane potential activity $V$ is shown in the blue traces on top. The red horizontal dashed lines indicate the peak amplitude of the currents at control temperature (10°C) to visualize whether it increases or decreases at 25°C. This is indicated by the arrows in each row (in colors: ⇑ red increases, ⇓ green decreases). In this example, the *Na* current increases its amplitude by more than two fold over the 15°C range. Despite the fact that $gCaT$ increases with temperature, the

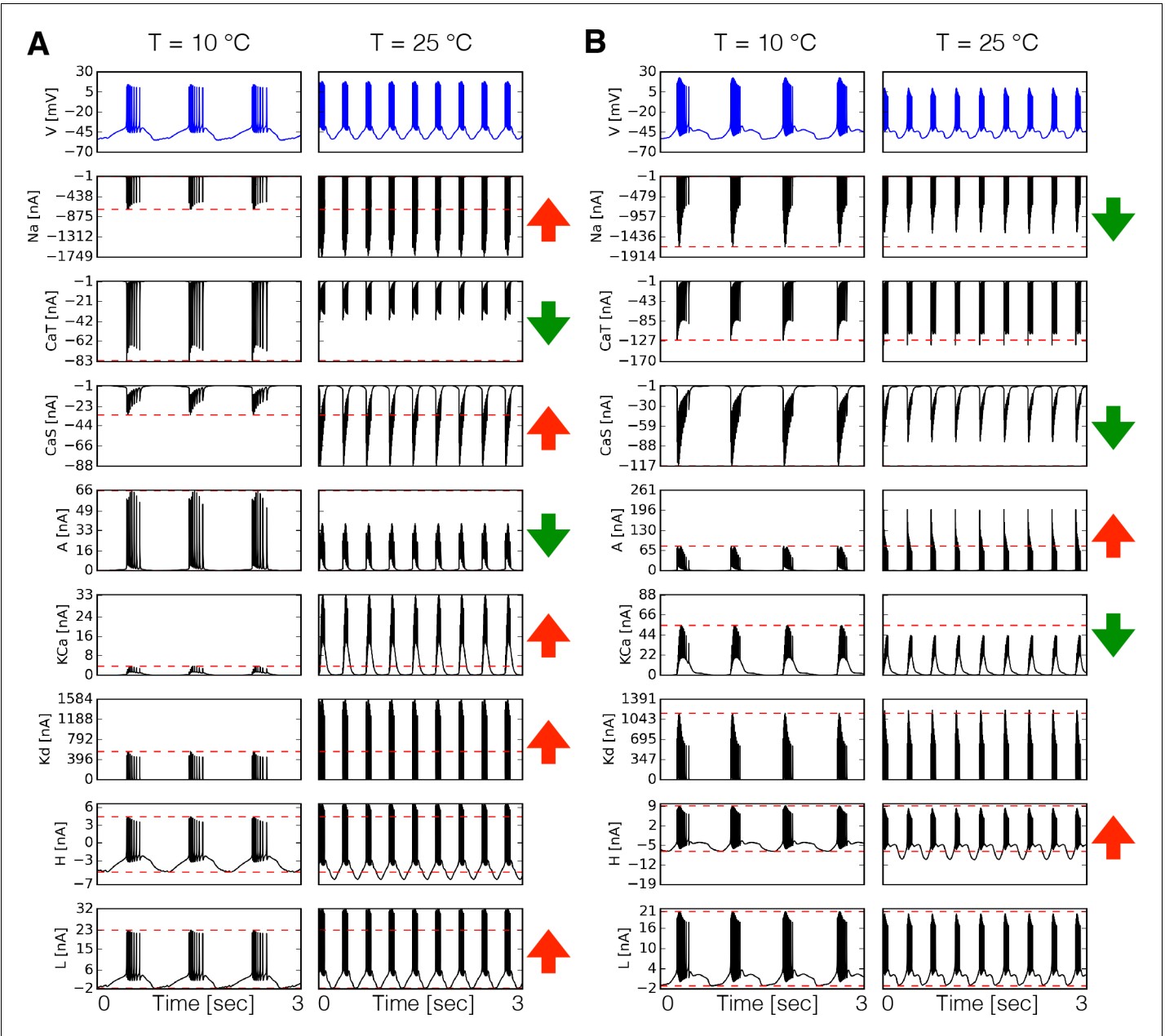

**Figure 3.** Dynamics of the currents at two temperatures. Currents of the *PD* cell for two different models (**A-B**). Temperature increases the peak amplitudes of some currents and it decreases it for others, as indicated by the ⇑ and ⇓ arrows in red and green. The way these currents are modified can vary substantially across models.

amplitude of the current decreases to about half its control value at 10°C. Similar observations can be made for other currents (*Figure 3A*). Note that all $Q_{10}$ values in these models are greater than 1. Therefore, when the current decreases it is because the relative timing of the currents' activation is changed by the neurons' firing properties. The way currents are modified by temperature is different across models. *Figure 3B* shows the currents in the PD cell at two temperatures for another model. In this case, the *Na* current decreases by a small amount despite the fact that the maximal conductance and timescales of this current increase with temperature. The *KCa* current which in *Figure 3A* grows by a factor of six, is moderately decreased by temperature in *Figure 3B*.

Because the total number of currents in the circuits is 31, it becomes cumbersome to compare them separately across temperatures (and models). For this reason, we computed and inspected their currentscapes (*Alonso and Marder, 2019*). The currentscapes use colors to show the percent contribution of each current to the total inward (or outward) current and are useful to display the dynamics of the currents in a compact fashion. *Figure 4* shows the currentscapes of each cell in one model at 10°C and 25°C. The shares of each current of the total inward and outward currents are displayed in colors over time. The total inward and outward currents are represented by the filled black curves on a logarithmic scale at the top and bottom. The currentscapes are noticeably different at 10°C and 25°C indicating that the currents contribute differentially across temperatures. For example, in the *PD* cell the *Leak* current contributes a visible share of the inward current both at the beginning and the end of the burst at 10°C, but at 25°C its contribution is mainly confined to the

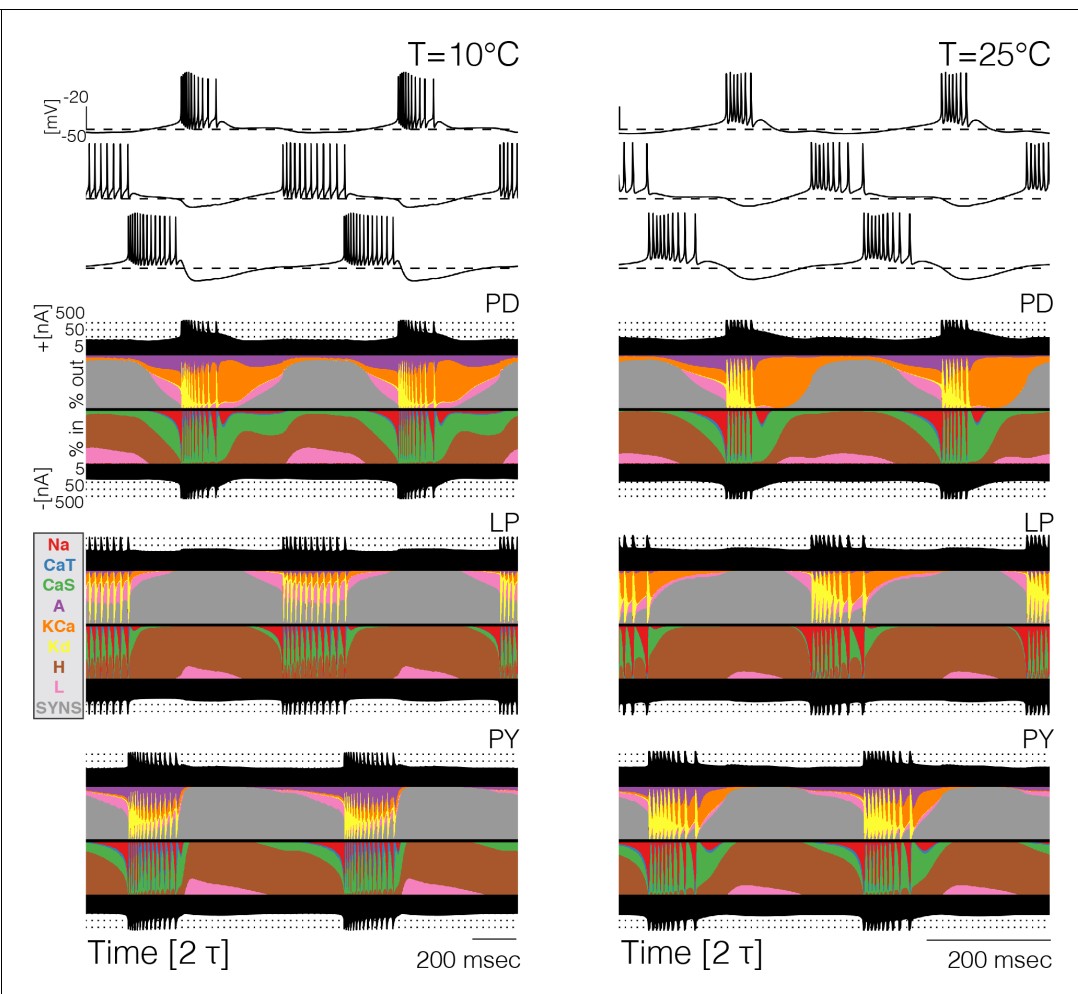

**Figure 4.** Currentscapes at different temperatures. The currentscapes use colors to display the percent contribution of each current type to the total inward and outward currents over time. The filled lines at the top and bottom indicate the total inward/outward currents in logarithmic scale. The panel shows the currentscapes for the three cells in the model at two temperatures. The panels show 2 periods $\tau$ of the oscillation.

beginning of the burst. The $A$ current is evenly distributed during the burst at 10°C but at 25°C its contribution is visibly larger at the beginning of the burst. In all cells, but most notably in the $LP$ and $PY$ neurons in this example model, temperature substantially increases the contribution of the $KCa$ current toward the end of the bursts.

The currentscapes show that in all models and at all temperatures, there are periods of time when the activity is dominated by one or two currents, and periods of time when several currents contribute to it by similar amounts. During spike production, the activity is first driven by $I_{Na}$ and later by $I_{Kd}$. During these periods, the total inward and outward currents differ by orders of magnitude and are entirely dominated by one current type. Similarly, when the cells are most hyperpolarized the dominant currents are $I_H$ and the synaptic currents $I_{Syn}$. There are periods of time—such as the beginnings and ends of bursts—when the total inward and outward currents are composed of comparable contributions of several current types. It is during these periods, when multiple currents act together to change the membrane potential, where we observe the most dramatic changes with temperature.

*Figure 5* shows 20 ms after the burst in a $PD$ neuron over temperatures between 10°C and 25°C. Almost all of the currents change over this temperature range, most notably there is almost a trade-off between the contributions of the $A$ and $KCa$ currents, with $A$ decreasing smoothly and $KCa$ increasing smoothly. The specifics of how the contributions change is different across models, but in all cases there is a smooth transition between different relative fractions of currents. This allows the neuron to slide smoothly through changes in burst termination mechanisms.

Different compensation mechanisms fail in different ways. *Figure 6* shows full currentscapes for two network models as they 'crash' at high temperature. Note that the complete loss of activity in the $PD$ neurons results from different currents being active in the two examples. Further examples of high temperature 'crashes' are shown in *Figure 6—figure supplement 1*. These examples show that the networks are crashing by different mechanisms in each case.

## High temperatures produce disordered circuits or 'crashes'

For temperatures higher than a critical temperature, the biological network displays anomalous regimes characterized by the emergence of slow time scales and intermittency between what appear to be metastable states (*Tang et al., 2012*; *Haddad and Marder, 2018*). At temperatures higher than 25°C some models do not display irregular behavior but just become quiescent, or produce subthreshold oscillations. Other models display irregular and seemingly aperiodic regimes. As expected, irregular regimes typically take place at values of the temperature near a transition between qualitatively different patterns of activity. *Figure 7* shows the activity of five models at

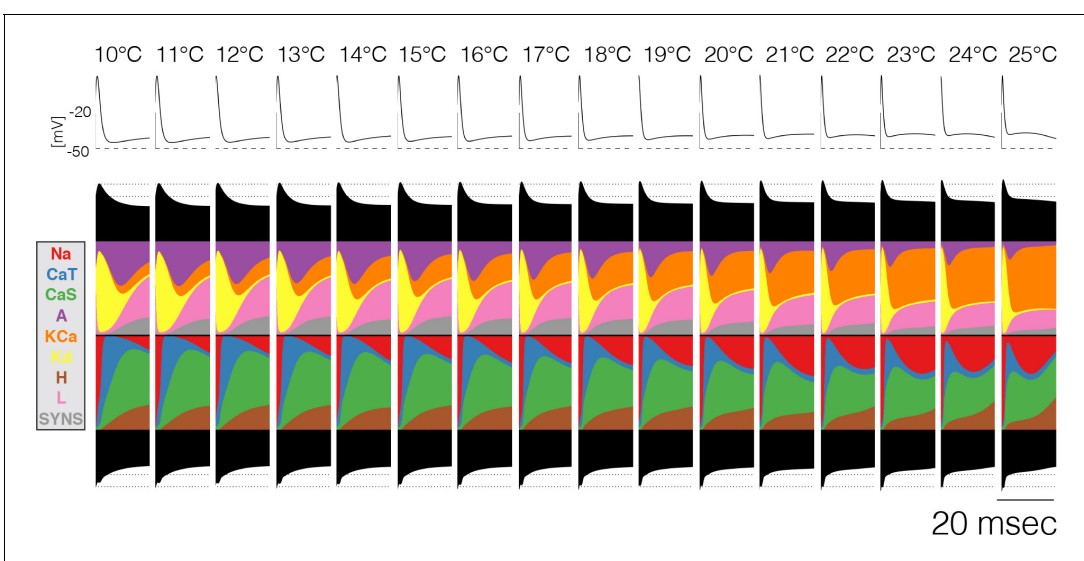

**Figure 5.** Current contributions at the end of bursts across temperature. The figure shows the currentscapes of the PD cell for the 20 ms following burst termination, and how these change over temperature (same model as in *Figure 7A*).

temperatures higher than their working regimes. The left panels show 30 s of data and the right panels show 5 s. In all cases, there are timescales in the behavior that are much longer than the period of the rhythm at 10℃ ($\approx$ 1 s). The dynamics of the models in these regimes is daunting and their characterization is beyond the scope of this work. However, qualitative features of these states such as the appearance of slow timescales or absence/presence of activity in one or more cells, are captured by these models. This, together with the multistable regimes shown before plus a source of noise, may provide a reasonable model to account for the irregular regimes in the biological network (*Tang et al., 2012*; *Haddad and Marder, 2018*).

## Hysteresis and multistability

It is natural to ask whether these models show hysteresis when the temperature is returned to control levels. Moreover, multistability is also often observed in conductance-based models of neuronal activity and biological neurons (*Cymbalyuk and Shilnikov, 2005*; *Malashchenko et al., 2011*; *Marin et al., 2013*; *Lechner et al., 1996*; *Paydarfar et al., 2006*), and the network studied here also exhibits such features. We explored hysteresis in the models by increasing temperature linearly from 10℃ to 35℃ in 30 min and then decreasing it back to 10℃ at the same rate. *Figure 8* shows the spiking patterns for two *PD* neurons during such temperature ramps (schematized on top). In *Figure 8A* the ISI distributions show bursting activity up to a critical temperature at which the model becomes quiescent. The quiescent state remains stable as temperature continues to increase and eventually recovers activity during the decreasing ramp, but at a lower temperature than when it stopped spiking during the increasing ramp. This implies that there is a range of temperatures where the model can be either quiescent or spiking, and thus is multistable. In most models (34 of 36), the spiking patterns over the working temperature range are similar across the up and down ramps but are not identical, suggesting there are multiple stable attractors that correspond to the triphasic rhythm but differ slightly in their spiking patterns (*Figure 1—figure supplement 2E*). This is consistent with the findings in *Tang et al., 2010* where temperature was ramped slowly and no significant differences were found when ramping up and down in temperature.

We also found models with pathological stable attractors that coexist with the pyloric rhythm at working temperatures. *Figure 8B* shows a model in which hysteresis is salient. At high temperatures (times between 25 and 35 min), this model produces *ISI* values as long as $\approx$ 1 s and remains spiking at all temperatures. The spiking patterns differ noticeably on the up and down ramps. The spiking pattern of the *PD* cell changes from regular bursting to a pathological state that remains stable until the model is ramped back down to $\approx$ 20℃. In our simulations, hysteresis mainly occurs at high temperatures indicating that states such as those in *Figure 7* can be multistable. The extent such hysteresis can be revealed in the biological experiments depends on the details of the perturbation protocol, and its currently under study.

## Response to perturbations at different temperatures

In all models, the dynamics of the currents are different at different temperatures and their contributions become reorganized in complex ways. The combined activity of all the variables in the model—the dynamical attractor—is therefore expected to have different stability properties at different temperatures. This means that an extreme perturbation can have qualitatively different effects at different temperatures. The responses of different models to extreme perturbations such as partially or completely removing a current can be diverse at any temperature (*Alonso and Marder, 2019*). While this is expected due to the complexity of these models, the fact that these responses can also be qualitatively different across temperatures indicates that changes in the current contributions are meaningful and affect the stability of the network. To shed light on this issue, we performed several simple perturbations.

*Figure 9* shows one perturbation in two models. In *Figure 9A*, we explored the interaction between temperature and the effect of removing the *A*-type K$^+$ channel at two temperatures: the models were simulated for 20 s in control conditions and then $I_A$ was completely removed for the next 20 s. *Figure 9A* (top) shows that removing the *A* current at 10℃ has catastrophic consequences for the network activity which becomes irregular. When the same perturbation is performed at 25℃ (bottom), the triphasic rhythm is almost normal except for the irregular bursting of *PY*. *Figure 9B* shows a similar pertubation in which we remove the *KCa* current. This model displayed normal

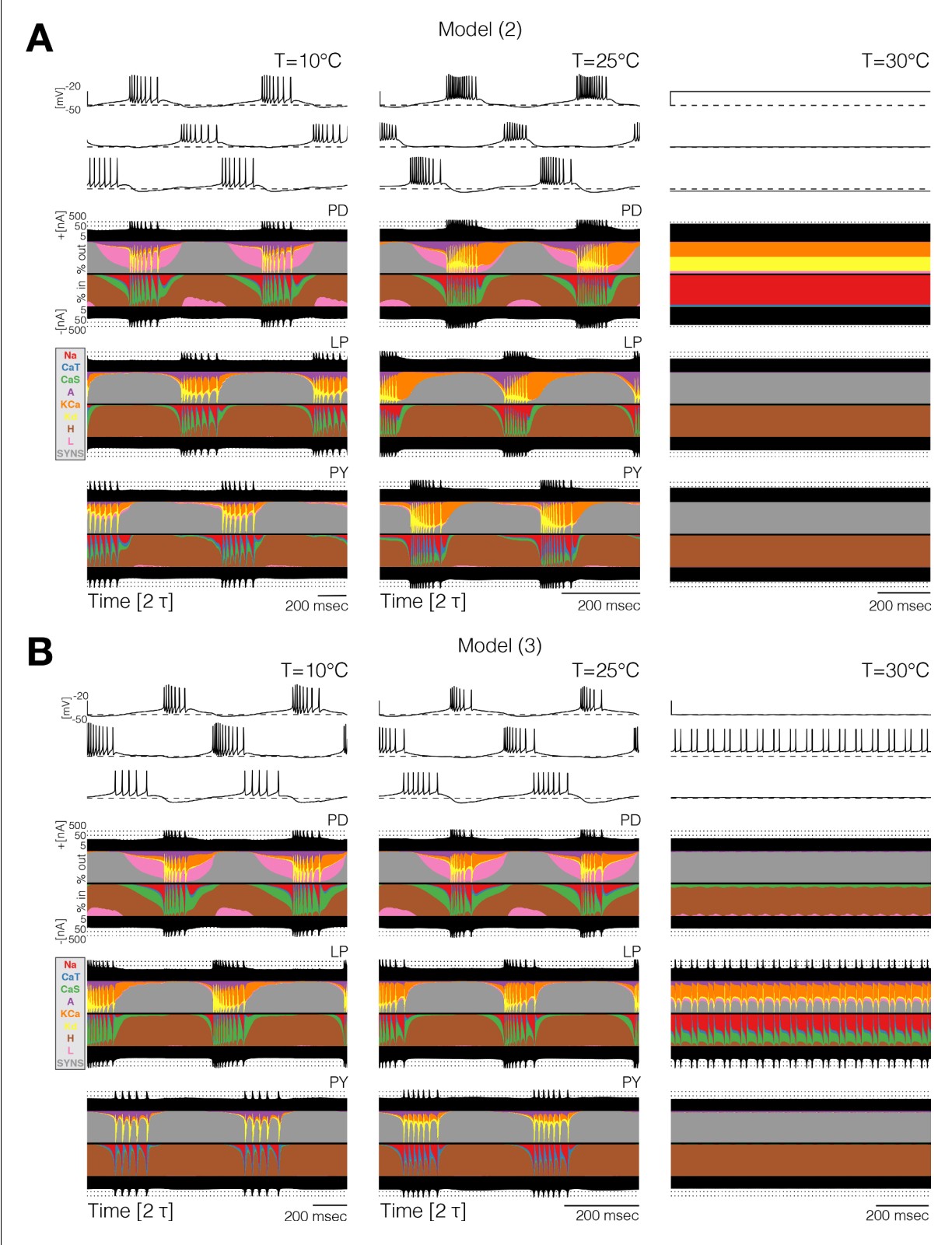

**Figure 6.** Currentscapes reveal that 'crashes' occur by different mechanisms. Currentscapes of two networks at 10°C and 25°C showing phase compensation, and 'crashes' at 30°C.

The online version of this article includes the following figure supplement(s) for figure 6:

**Figure supplement 1.** Currentscapes at crashed states.

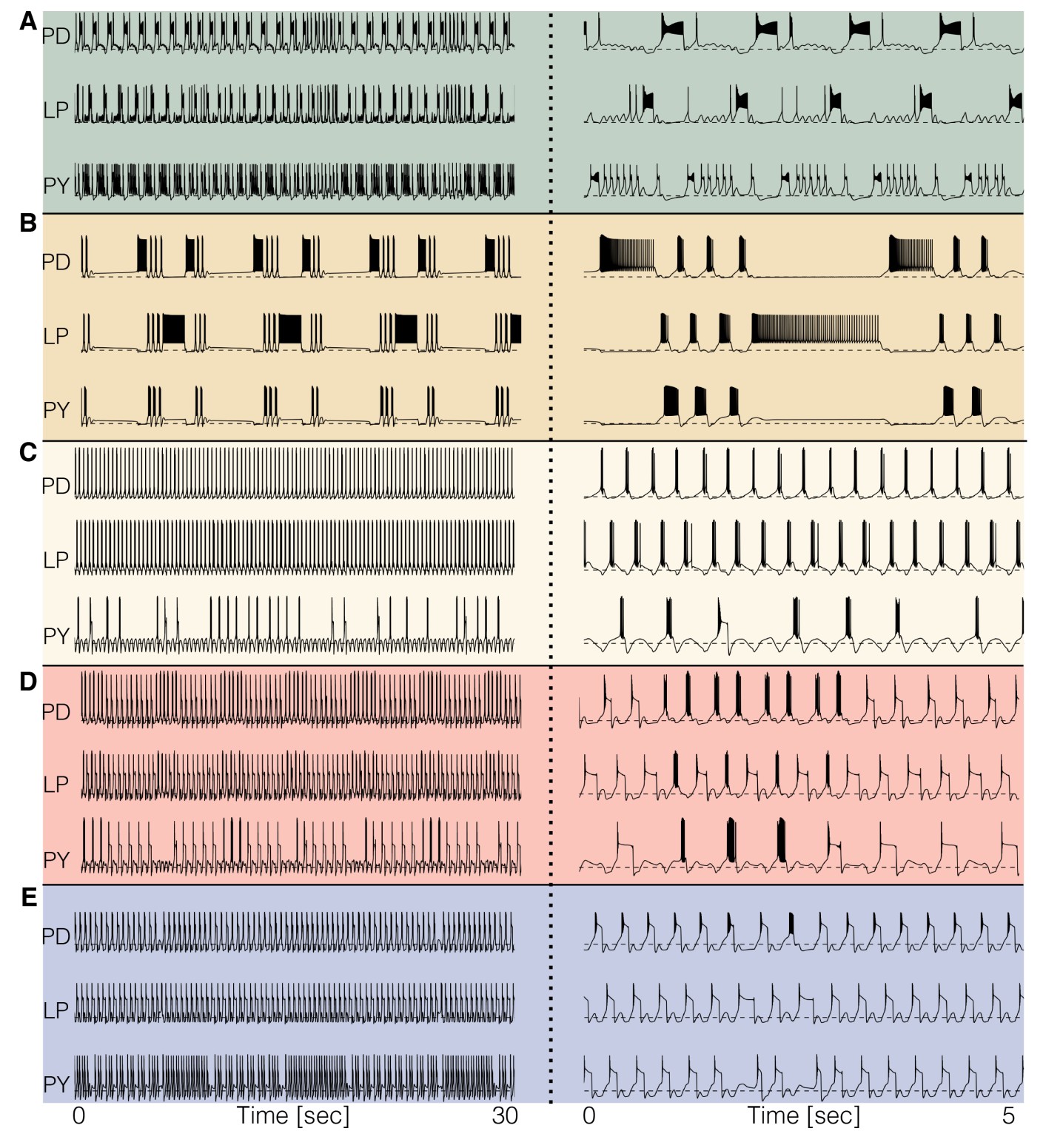

**Figure 7.** Disordered circuits at high temperatures. The models become dysfunctional—or crash—at temperatures higher than 25°C and do so in different ways. (A-E) Membrane potential of five models. The traces on the *left* show 30 s of data and the traces on the *right* show an expanded trace of 5 s. Note the emergence of time scales much longer than that of the original rhythm. Temperatures in the simulations are (A–E): 34°C, 34°C, 33°C, 29°C, 30°C.

triphasic behavior at 10℃ (top) and when the $KCa$ current was removed its activity slowed but the triphasic rhythm was maintained. The same perturbation at 25℃ results in quiescence. In this model, the $KCa$ current is necessary for the activity at 25℃ but not at 10℃.

To further characterize the differential response to perturbations at different temperatures, we performed a second assay in which we gradually decreased a current from control (100%) to complete deletion (0%) in five steps. We performed these simulations for all 36 models and all current types at 10℃ and 25℃ and compared their responses. In general, gradually removing a current can result in qualitatively different states depending on the temperature at which this perturbation is performed. The responses are so diverse that a coarse classification scheme is sufficient to highlight this observation. We classified the network activity based on the spiking patterns and waveforms of each cell, and also by their relative activities. The activities of each cell were classified as: regular bursting, irregular bursting, tonic spiking, irregular spiking, single spike bursting, quiescent and other, while the network activity was classified as triphasic or not triphasic. The classification scheme consists of a simple decision tree based on the statistics of spiking and is sufficient to tease apart the different regimes we observe at a coarse level (Materials and methods).

*Figure 10A* shows the result of classifying the responses of one model to gradually decreasing a conductance in five steps, at two different temperatures, for four conductances. We employed 5 × 4 *response grids* to summarize the effects of partially or completely removing a conductance at 10℃ and 25℃. The rows in the grids correspond to different values of conductance removals, with the control condition (100%) on top, and the completely removed condition (0%) at the bottom. The first three columns correspond to each of the cells (PD-LP-PY) and the fourth column corresponds to the network (NET). The colors indicate the type of activity of each cell and the network color coded for condition as indicated by the labels in the figure. The top left panel in *Figure 10A* shows the responses of one model to removing the sodium current $I_{Na}$. At 10℃, when $gNa \rightarrow 75\% gNa$ the $LP$ cell bursts irregularly and the activity of the network is *not triphasic*, but if the same perturbation is performed at 25℃ the activity remains triphasic. Removing the $KCa$ current in this model has little effect at 10℃ as the activity remains triphasic. However, this current plays an important role at 25℃ and the model becomes quiescent upon complete removal of $I_{KCa}$. The same observation that the responses to perturbation are different at each temperature holds regardless of the perturbation type. In the cases of the $CaT$ and $CaS$ currents complete removal results in different non-triphasic activity at each temperature. The responses of the model are in general qualitatively different at different temperatures, and they are also diverse across models (not shown). The fact that the models 'break-down' in different ways is consistent with the observation that these network mechanisms are diverse. In all models, the responses to extreme perturbation are temperature dependent for most perturbation types. In *Figure 10B* we quantify how many models respond differently to complete removal of a current at 10℃ and 25℃. This corresponds to comparing the bottom row of the response grids at each temperature, for each conductance type. In most cases, about half of the models respond to complete removal in different ways at each temperature. The exceptions are the $CaT$ current for which most models respond in different ways, and the $H$ current where complete removal results in equivalent states (quiescence in this case) at both temperatures.

## Discussion

Enzymes and ion channels show altered behavior when temperature is changed, as temperature influences their conformation. Yet many animals live successfully over wide temperature ranges. For example, North Atlantic lobsters and crabs routinely experience more than 25℃ temperature swings during a year, and may see 8 - 10℃ changes within a short time period (*Haefner, 1977*; *Stehlik et al., 1991*). This raises a series of fascinating questions of how behavior and circuit performance can be maintained while all biological molecules are, to a greater or lesser degree, differentially temperature dependent (*Robertson and Money, 2012*).

It is important to distinguish between altered activity that occurs rapidly in response to acute, or immediate changes in temperature and those changes that occur during slow adaptation and acclimation, processes that can take days, weeks or months. In this study, we have modeled the fast changes that we assume would occur 'automatically' as proteins change conformation and properties rapidly, and as are usually measured in their $Q_{10}$'s. This is to be contrasted with the slower, long-term effects that result in acclimation in which a variety of molecular processes are triggered that

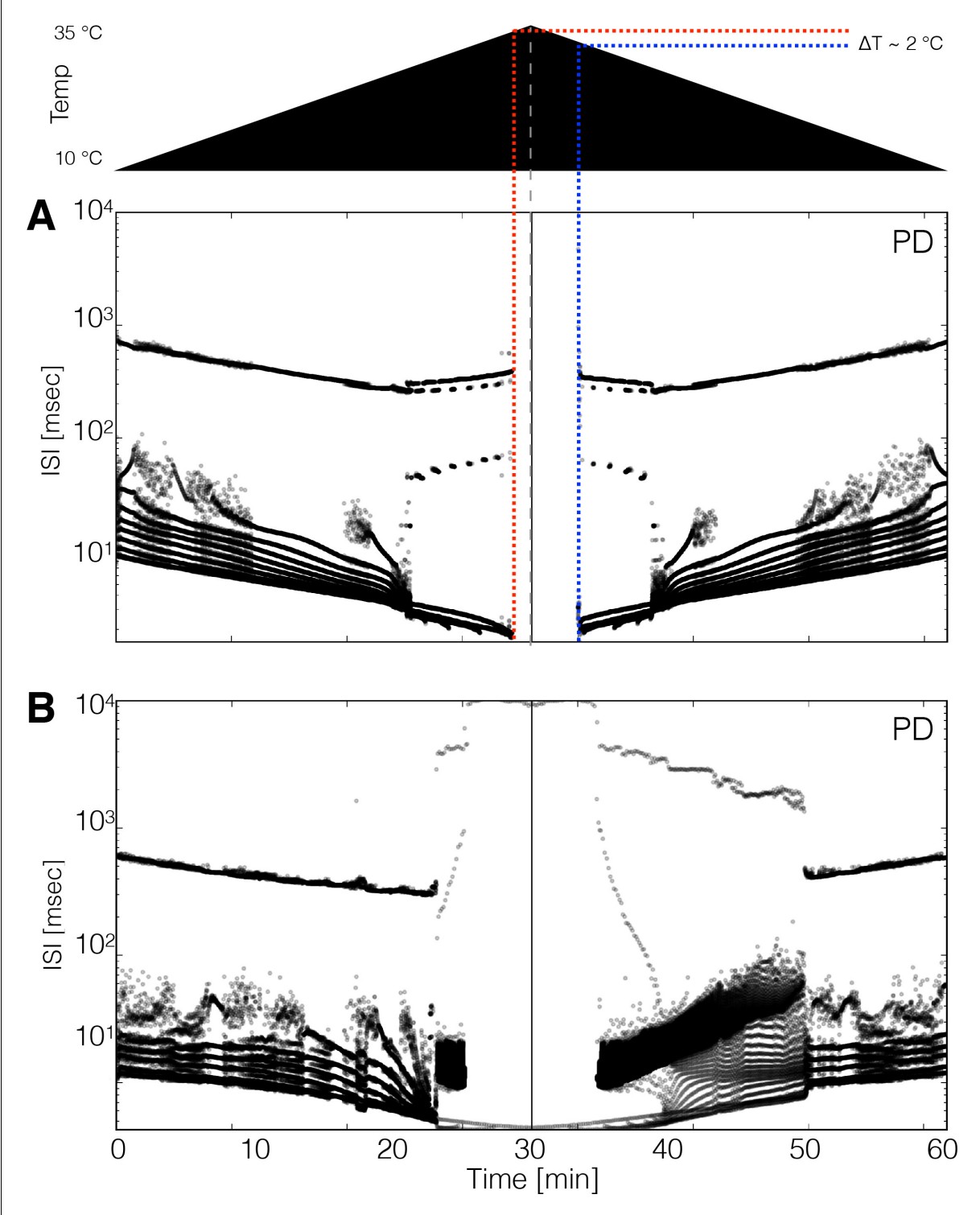

**Figure 8.** Hysteresis and multistability. Temperature was increased from 10°C to 35°C and then decreased back to 10°C symmetrically. (Top) Schematic representation of the temperature ramp. Panels **A** and **B** show the ISI distributions of the *PD* cell over time for two different models. (**A**) The cell ceases to produce spikes before 600 s, which is also before reaching 35°C, and remains quiescent until temperature is ramped down. The temperature at which the cell resumes spiking is ≈ 2°C lower than the temperature at which it became quiescent, indicating that the system is multi-stable over that temperature range. (**B**) Hysteresis is more evident in this model. The spiking patterns during the down ramp differ visibly from those during the up ramp over a wide temperature range.

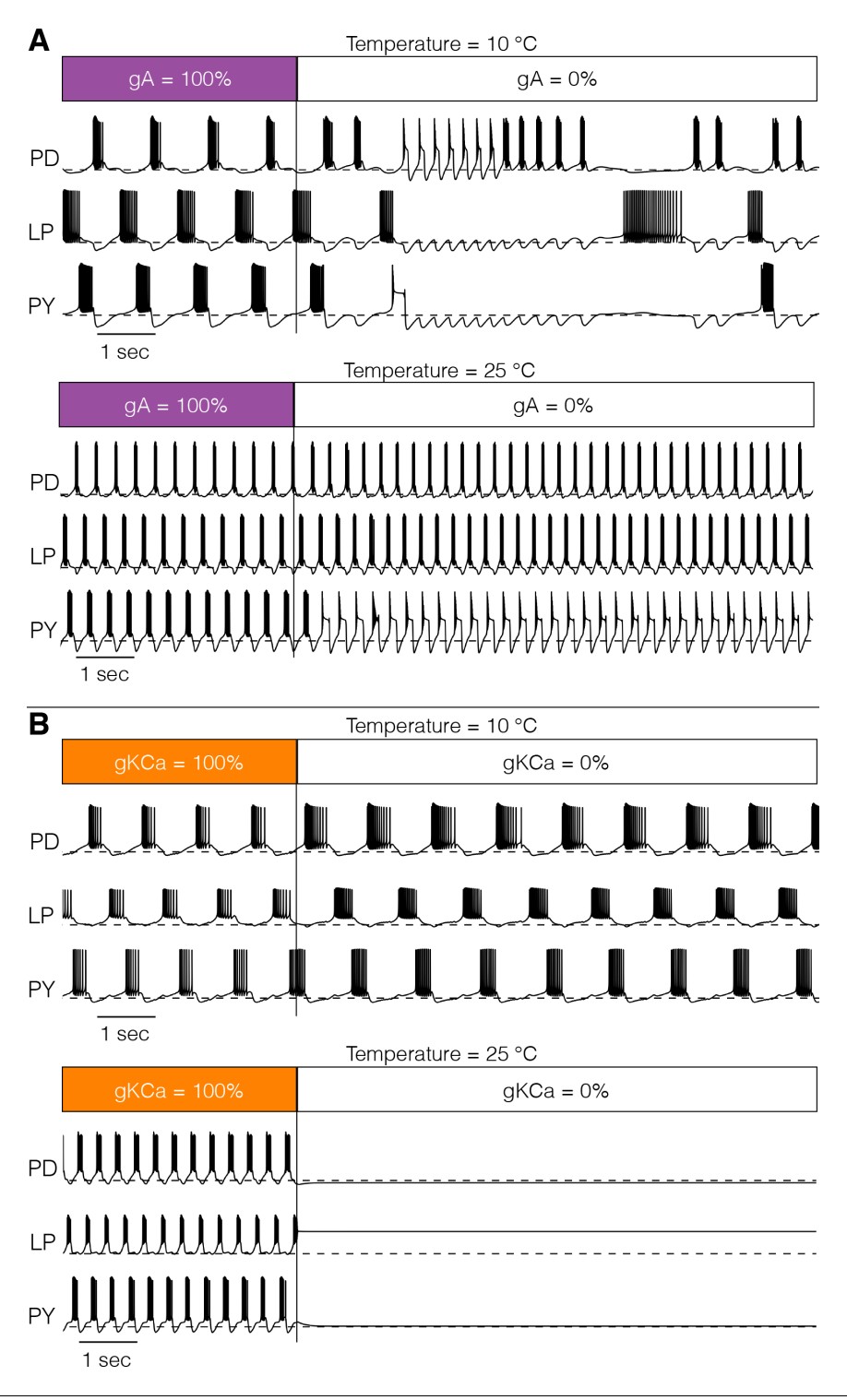

**Figure 9.** Response to perturbations at different temperatures. The response of the models to extreme perturbations can be different at different temperatures. The figure shows the responses of models to the same perturbation at two different temperatures. (**A**) (top) Membrane potential over time at 10°C. The first 4 s correspond to the control condition with all currents intact. At $T = 4sec$ we removed the $A$ current. (bottom) Same perturbation performed at 25°C (same model as in *Figure 4*). (**B**) Same protocol as in A but removing the $KCa$ current in a different model (*Figure 7A*).

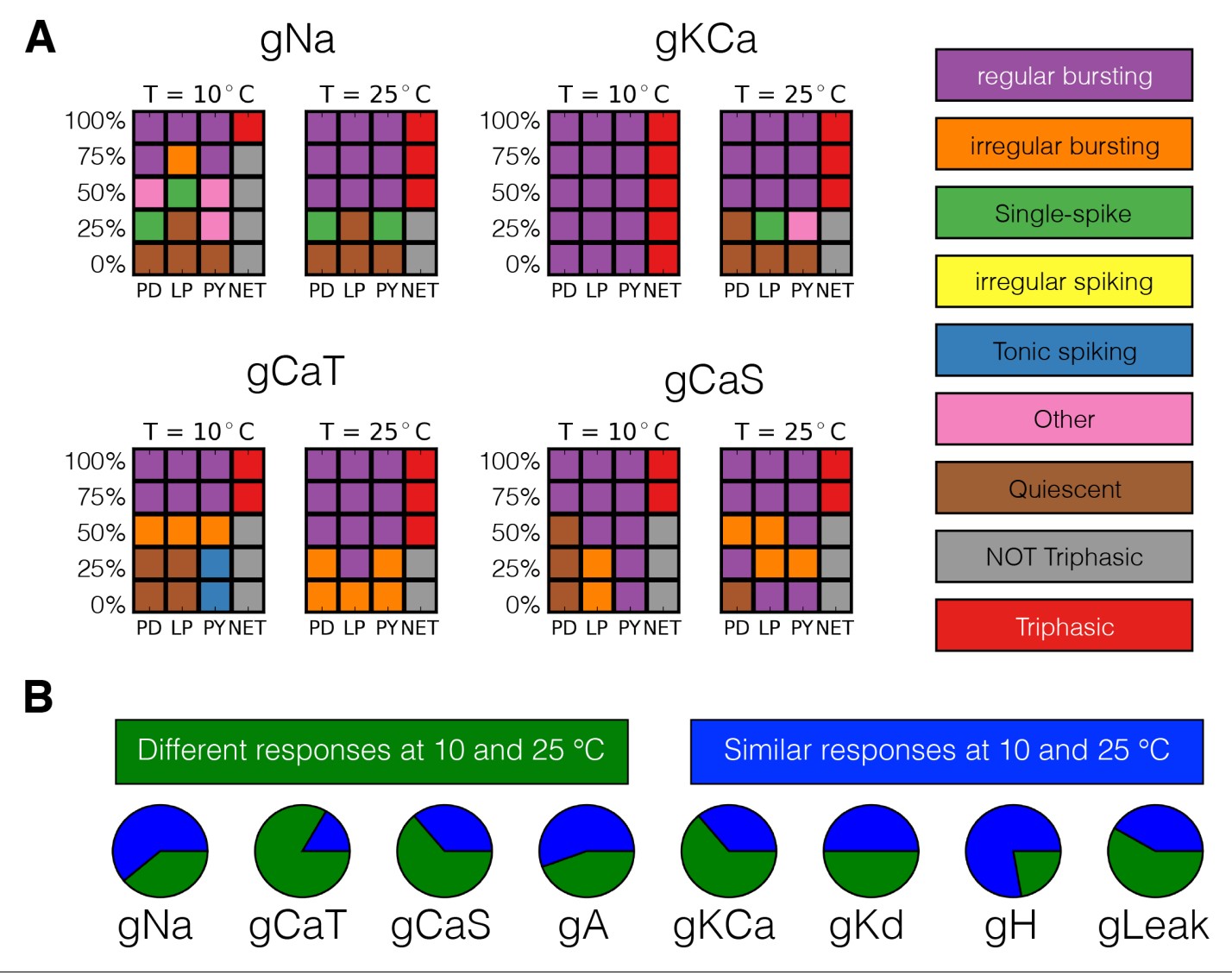

**Figure 10.** Classification of responses at different temperatures. The models' responses to perturbations are diverse and are in general different at different temperatures. We classified the responses of the models at two temperatures for five values of gradual decrements of each current. (**A**) Response to removal of *Na*, *KCa*, *CaT* and *CaS* currents for one model. (**B**) The pie charts show the number of models for which the responses to complete removal (0%) are classified as the same (green) or different (blue) at 10℃ and 25℃.

may in turn change the proteins present in the membrane. These mechanisms, and much adaptation to long-term changes in temperature depend on the activation of heat shock proteins, RNA editing and a whole array of molecular mechanisms (*Garrett and Rosenthal, 2012*; *Rosenthal, 2015*; *Martin Anduaga et al., 2019*). We will first discuss the issues that arise from the examination of robustness to acute temperature changes, as modeled here, and then offer some speculations for additional processes that may be important to enhance robustness in response to long-term temperature changes.

Unfortunately, there are relatively few data relevant to whether the $Q_{10}$ of a channel in a given animal changes in response to acclimation or seasonal conditions. Therefore, although it is often assumed that a $Q_{10}$ for a given protein is invariant over the lifetime of the animal this may not necessarily be the case. Another difficulty is that there are few, if any, studies of many ion channel $Q_{10}$s in the same animal in the same temperature range. Computational modeling will not answer what the 'real' conductance and $Q_{10}$ combinations are.

The assumption that these quantities are stationary throughout the life of an animal (and/or the same across species) may not be warranted. For example, long-term acclimation over months resulting in RNA editing, could alter some $Q_{10}$s. Moreover, it is possible that minor genetic variations in channel proteins exist in the population so that different animals in the population could show modestly different $Q_{10}$s. Therefore, the point of this study is to ask generically how compensation occurs acutely assuming different temperature sensitivities for the processes, within the experimental range. For these reasons, in the absence of data, it is valuable to explore the $Q_{10}$ space to have on overview of how compensation takes place acutely at any point during the lifetime of an animal.

One of the most important and non-intuitive results from this work is seen in *Figure 5*, which demonstrates that a neuron can smoothly slide through mechanisms that govern a physiological process as temperature is changed. In the case of *Figure 5*, the repolarization of the last spike in the burst shows a large contribution from $I_A$ at 10°C but becomes small at 25°C while $I_{KCa}$ contributes little at 10°C and becomes dominant at 25°C. This shows that while both of these $K$ currents can contribute to spike and burst repolarization, their shares in these processes change, and the neuron is robust as it smoothly slides through these mechanisms. This smooth transition can occur precisely because these three K$^+$ currents depend on voltage, time, and temperature differently.

We speculate that one advantage conferred by the expression of many different ion channels that differ in their properties is to provide many sets of such 'sliding stability mechanisms'. Individual neurons in all species have large numbers of ion channel genes that encode proteins that give rise to the membrane currents that are usually measured and characterized in voltage-clamp. The model neurons used are loosely based on voltage-clamp measurements from crab and lobster stomatogastric ganglion (STG) neurons (*Golowasch et al., 1992*; *Turrigiano et al., 1995*) and have eight currents. That said, we now know that there are at least twenty to thirty and probably more voltage-gated ion channel genes in individual STG neurons (*Northcutt et al., 2016*). In some cases, several of these genes may contribute to one of the currents that we model here. For example, the delayed rectifier current modeled here may have contributions from at least three different genes, and $I_A$, modeled here, also probably has contributions from at least two different genes. Given that these genes encode similar but not identical protein subunits, it is not unreasonable to expect that these proteins may respond differently to temperature, and that the actual composition of the activated $K$ currents may reflect a change in these contributions as a function of temperature. This again, could provide still another 'sliding stability mechanism', and it would be interesting in the future to create models using data from currents that have been measured in isolation, such as in *Ranjan et al., 2019*.

The same principle of sliding circuit mechanisms is likely to hold when we approach network dynamics. The phase relationships of the pyloric rhythm are maintained over a considerable temperature range (*Tang et al., 2010*; *Soofi et al., 2014*). Likewise, the pyloric phase relationships are also maintained over a considerable frequency range (*Hamood et al., 2015*). In this latter case, it has been argued that the phase compensation depends on the conjoint action of a number of different cellular mechanisms, including synaptic depression (*Manor et al., 1997*; *Hooper, 1997*), and the activation and inactivation of $I_A$ and $I_H$ (*Nadim et al., 1999*; *Nadim and Manor, 2000*; *Tang et al., 2010*; *Harris-Warrick et al., 1995*). So, here the principle also holds: resilience and robust function may require smoothly moving between a variety of different cellular mechanisms.

Long-term mechanisms for temperature adaptation and acclimation may extend robustness by changing the functional properties of ion channel proteins. In biological systems, sustained exposure to temperature often activates heat shock proteins and has effects on many cellular and molecular functions (*Sharp et al., 1999*). Interestingly, changes in ion channel composition can result from RNA editing, or changes in splice variants (*Johnson et al., 2011*; *Lin and Baines, 2015*). It is easy to imagine that these kinds of molecular mechanisms can result in shifts in the temperatures at which neurons and circuits may function. Acclimation to high temperatures usually requires at least 3–4 weeks of sustained alteration of the environmental temperature, and this is enough time to replace a large subset of ion channel proteins in the membrane. This raises a series of fascinating questions for how homeostatic regulation of excitability may be preserved if the molecular mechanisms that are engaged in homeostasis are also influenced by temperature.

Hysteresis in response to perturbation can result from a variety of time-dependent mechanisms. *Figure 8* shows examples of hysteresis that can occur in response to temperature ramps, even in models in which the effects of temperature on the properties of each ion channel are viewed as

essentially instantaneous. This hysteresis can occur because at high temperatures (> 25°C) the network can produce several different patterns of activity or dynamical states (multistability). The activation and inactivation rates that characterize ion channel function are different for each of these states and therefore, as temperature is lowered, they lose stability in different ways. Of course, any long-term molecular mechanisms (not modeled here) that alter channel expression or splicing or phosphorylation, could produce long-lasting hysteresis.

In this work, we used a genetic algorithm (*Alonso and Marder, 2019*) to find temperature robust networks. It is important to reiterate that temperature robust neurons and networks are difficult to find doing random searches (*Caplan et al., 2014*). Therefore, despite the significant animal-to-animal differences in conductance densities seen across the population, randomly sampling conductance values is unlikely to result in successful solutions for individual animals. This argues that evolution has arrived at a series of biological mechanisms that give rise to successful temperature compensation. These are likely to include rules by which correlated values of conductances are produced, such as those found in homeostatic models (*O'Leary and Marder, 2016*; *O'Leary et al., 2013*).

Taken at face value, models suggest that conductance densities drift throughout the life of an animal as a result of homeostatic processes (*LeMasson et al., 1993*; *Liu et al., 1998*; *Golowasch et al., 1999a*; *O'Leary et al., 2014*). Conductance densities can also change in an activity-dependent manner as a result of perturbations (*Turrigiano et al., 1994*; *Golowasch et al., 1999a*; *Golowasch et al., 1999b*; *Golowasch et al., 1999a*; *Santin and Schulz, 2019*; *Golowasch, 2019*). The temperature sensitivities of ion channels are dictated by their molecular structures, and for this reason it is tempting to assume that the temperature sensitivity of a given channel is similar throughout the lifetime of the animal. Nonetheless, as all channels are subject to splice variants and other mechanisms of molecular regulation, it is possible that environmental temperature results in changes in $Q_{10}$ that are consistent with acute robustness, even if there have been changes in conductance densities. This suggests that conductance densities and $Q_{10}$ regulation may be coordinately controlled in biological systems.

One of the most significant results of this paper is that temperature robustness of network function can take place in multiple different ways. In these models, both the membrane potential and the spiking patterns are affected by temperature changes and these changes appear to be different in every model we inspected. Across the models, different values of the maximal conductances G and temperature sensitivities $Q_{10}$ result in consistent differences in the duty cycle distributions. Measurements of the values of the maximal conductances in the STG show large variability across individual animals (*Goaillard et al., 2009*; *Schulz et al., 2006*; *Schulz et al., 2007*; *Temporal et al., 2014*; *Northcutt et al., 2016*) so our expectation is that the duty cycle distributions of the biological cells will also display intricate dependencies with temperature, and that these distributions will be different across individuals.

Temperature is not the only perturbation that crabs and lobsters experience. As with temperature, the responses of the STG to changes in pH are diverse across individuals (*Haley et al., 2018*), again consistent with the large amount of animal-to-animal variability in the expression of ion channels (*Schulz et al., 2006*; *Temporal et al., 2014*; *Tran et al., 2019*). It is therefore reasonable to assume that the responses to a global perturbation are diverse across individuals because different compositions of channel densities—which produce similar pyloric rhythms—are differentially resilient to any given perturbation. By the same token, as currents change as a function of temperature we expect that a second global perturbation of an individual may have qualitatively different effects at different temperatures, as is illustrated with the models in *Figure 10*. The interaction between temperature and a second perturbation is the subject of recent experimental studies (*Haddad and Marder, 2018*; *Ratliff et al., 2018*). These studies are consistent with the interpretation that different current configurations have different stability properties and that temperature changes these configurations.

One of the general take-home lessons from this work is that models can be resistant to a pharmacological perturbation at one temperature but the same model can be sensitive to the same perturbation at another temperature (*Figure 10*), and that different individuals may differ in their responses to the same perturbation. The potential implications of this for therapeutics are evident. But also evident, is that the search for therapeutic agents can be fraught if the assay itself is differentially sensitive to temperature or another perturbation. Moreover, the 'sliding mechanisms' provided

by multiple ion channel genes and their molecular regulation suggests that over time therapeutic sensitivity to a given pharmacological intervention could be quite variable. These findings may contribute to our understanding of individual variable therapeutic efficacies within the population and across time within an individual.

## Materials and methods

### The model

The activity of the cells was modeled using single-compartment models similar to those described previously (*Turrigiano et al., 1995*; *Liu et al., 1998*; *Goldman et al., 2001*; *Alonso and Marder, 2019*). Each neuron has a sodium current, $I_{Na}$; transient and slow calcium currents, $I_{CaT}$ and $I_{CaS}$; a transient potassium current, $I_A$; a calcium-dependent potassium current, $I_{KCa}$; a delayed rectifier potassium current, $I_{Kd}$; a hyperpolarization-activated inward current, $I_H$; and a leak current $I_{leak}$. The number of state variables per cell is 13. The units for voltage are $mV$, the conductances are expressed in $\mu S$ and currents in $nA$. The pyloric network consists of three cells with the same ion channel types but different conductance densities. The interactions in the network consist of seven chemical synapses and are similar to *Prinz et al., 2004*. The synaptic current is given by $I_s = g_s s(V_{post} - E_s)$, where $g_s$ is the synapse strength, $V_{post}$ is the membrane potential of the postsynaptic neuron and $E_s$ is the reversal potential of the synapse. The activation of a synapse $s(t)$ is given by

$$\frac{ds}{dt} = \frac{s_\infty(V_{pre}) - s}{\tau_r + \tau_s} \tag{1}$$

with,

$$s_\infty(V_{pre}) = \frac{1}{1 + exp((V_{th} - V_{pre})/\Delta)}, \tag{2}$$

and

$$\tau_s = \frac{1 - s_\infty(V_{pre})}{k_-}. \tag{3}$$

These equations are identical to *Prinz et al., 2004* except for the inclusion of a bound for the timescale of activation $\tau_r = 20\,msec$ (we want to avoid the case that as $s_\infty \to 1$ then $\tau_s \to 0$ and $\dot{s} \to \infty$). All other parameters (except $g_s$) are identical to *Prinz et al., 2004*. Following *Prinz et al., 2004*, we set $E_s = -70\,mV$ and $k_- = \frac{1}{40}\,ms$ for glutamatergic synapses, and $E_s = -80\,mV$ and $k_- = \frac{1}{100}\,msec$ for cholinergic synapses. We set $V_{th} = -35\,mV$ and $\Delta = 5\,mV$ for both synapse types.

Temperature effects were included in this model as done previously by *Caplan et al., 2014* and *O'Leary and Marder, 2016*. Temperature dependence was introduced in the time constants of the channel-gating variables $\tau_{m_i}$ and $\tau_{h_i}$, the maximal conductances $g_i$, the time constants of calcium buffering $\tau_{Ca}$, and the maximal conductances and time constants of the synapses. This was done by replacing all conductances $g_i$ by $R_i(T)g_i$ and all time scales $\tau_i$ by $R_i(T)^{-1}\tau_i$ where

$$R_i(T) = Q_{10_i}^{\frac{T - T_{ref}}{10}}. \tag{4}$$

Here $T$ is the temperature, $T_{ref} = 10°$ is the reference temperature and $Q_{10_i}$ is defined as the fold change per 10°C from the reference temperature ($i$ indicates the process type). Finally, the calcium reversal potential $E_{Ca}$ depends on temperature through the Nernst equation.

The total dimension of the model is 46 = 3 × 13 + 7 with 13 state variables per cell, and 7 variables for the state of the synapses. In this work some parameters were considered fixed while others were allowed to take values in a range. The number of parameters we varied per cell is 9: the 8 maximal conductances plus the calcium time constant $\tau_{Ca}$. The temperature sensitivities were assumed to be the same for all cells and amount to 24 additional parameters. We need to specify 8 $Q_{10}$ values for the intrinsic maximal conductances (one per channel type), plus 11 $Q_{10}$ values for the time scales of each intrinsic process (not all currents inactivate), plus the $Q_{10}$ of the calcium time constant. The

number of $Q_{10}$ parameters for the synapses is 4:2 for the maximal conductances (of each synapse type), and 2 for the timescales of activation. The models were simulated using a Runge-Kutta order 4 (RK4) method with a time step of $dt = 0.05 \, msec$ (**Press et al., 1988**). We used the same set of initial conditions for all simulations in this work $V = -51 \, mV$, $m_i, h_i = 0$ and $[Ca^{+2}] = 5 \, \mu M$.

## Finding parameters

Each model is specified by the maximal conductances **G** and calcium time constants of each cell (9 × 3 parameters), and the temperature sensitivities **Q$_{10}$** of each process (24 parameters). We recently introduced a function that upon minimization, results in values of the maximal conductances for which the activity of a single compartment corresponds to periodic bursting with a target frequency ($f_{tg}$) and duty cycle ($dc_{tg}$). The function uses thresholds to obtain temporal information of the waveform, such as spike times, and then uses it to assign a score or error that measures how close the solutions are to a target activity. This function is described in detail in **Alonso and Marder, 2019** and was used here to find temperature robust networks.

For any given set of conductances we simulated the model for 20 s and dropped the first 10 s to minimize the effects of transient activity. We then computed the average ($<>$) burst frequency $<f_b>$, the average duty cycle $<dc>$, the number of crossings with a slow wave threshold $\#_{sw} = -50 \, mV$, the number of bursts $\#_b$, and the average lags between bursts $<\Delta_{PD-LP}>$ and $<\Delta_{PD-PY}>$. To discard unstable solutions we checked if the standard deviation of the burst frequency and duty cycle distributions was small; a solution was discarded if $std(\{f_b\}) \geq <f_b> \times 0.1$ or $std(\{dc\}) \geq <dc> \times 0.2$. If a solution is not discarded we can use these quantities to measure how close it is to a target behavior,

$$
\begin{aligned}
E_f &= \sum_{i=\text{cell}} (f_{tg} - <f_b>_i)^2 \\
E_{dc} &= \sum_{i=\text{cell}} (dc_{tg} - <dc>_i)^2 \\
E_{sw} &= \sum_{i=\text{cell}} (\#_{sw} - \#_b)^2 \\
E_{ph} &= (\Delta_{PD-LP}tg - \frac{<\Delta_{PD-LP}>}{\tau_b})^2 + (\Delta_{PD-PY}tg - \frac{<\Delta_{PD-PY}>}{\tau_b})^2 .
\end{aligned}
\tag{5}
$$

Here $E_f$ measures the mismatch of the bursting frequency of each cell with a target frequency $f_{tg}$ and $E_{dc}$ accounts for the duty cycle. $E_{sw}$ measures the difference between the number of bursts and the number of crossings with the slow wave threshold $t_{sw} = -50 \, mV$ (we ask that $\#_{sw} = \#_b$). $E_{ph}$ compares the lags between bursts (in units of the bursting period $\tau_b = \frac{1}{<f_b>}$) to a target lag $\Delta_{tg}$. These measures are discussed in more detail in **Alonso and Marder, 2019**.

Let **G** denote a set of maximal conductances (and $\tau_{Ca}$), we can then define an objective function

$$
E(\mathbf{G}) = \alpha E_f + \beta E_{dc} + \gamma E_{sw} + \eta E_{ph},
\tag{6}
$$

where the coefficients $(\alpha, \beta, \gamma, \eta)$ define how the different sources of penalties are weighted. In this work we used $\alpha = 10$, $\beta = 1000$, $\gamma = 1$, $\eta = 10$. These weights were found by trial and error and kept fixed. The penalties $E_i$ were calculated using $T = 10$ secs with $dt = 0.05$ msecs. The target control behavior was defined as having all cells bursting with 20% duty cycle for $PD$ ($dc_{tg_{PD}} = 0.2$) and 25% for the LP and PY cells ($dc_{tg_{LP,PY}} = 0.25$). The lag between bursts was targeted to be $\Delta_{PD-LP}tg = 0.5$ and $\Delta_{PD-PY}tg = 0.75$. The target burst frequency of all cells was set to $f_{tg} = 1 Hz$ (**Bucher et al., 2006**; **Tang et al., 2010**; **Hamood et al., 2015**).

Minimization of the objective function produces sets of maximal conductances **G** for which the resulting circuit activity mimics the pyloric rhythm. The minimization was performed over a search space of allowed values listed here: for each cell we searched for $g_{Na} \in [0, 10^3]$ ([$\mu S$]), $g_{CaT} \in [0, 10^2]$, $g_{CaS} \in [0, 10^2]$, $g_A \in [0, 10^2]$, $g_{KCa} \in [0, 10^3]$, $g_{Kd} \in [0, 10^2]$, $g_H \in [0, 10^2]$, $g_L \in [0, 10]$, $\tau_{Ca} \in [0, 2 \times 10^3]$ (msec). All synaptic conductances were searched in the range $g_{syn} \in [0, 5 \times 10^{-2}]$ ([$\mu S$]). The choice of search space contains our knowledge that some conductances like $gNa$ and $gKd$ are significantly larger than $g_{leak}$ or the synaptic conductances. This choice makes optimizations faster and produces diverse solutions. We discretized our search space by taking $10^3$ equally spaced values for each parameter. We minimized the objective function (**Equation 6**) using a custom genetic algorithm (**Holland, 1992**) on a desktop computer with 32 cores. For each network, we ran the algorithm using 1000 individuals taken at random in the search space for 10,000 generations. Finding a successful population takes several hours.

Temperature robustness is achieved in our models by searching sets of $\mathbf{Q_{10}}$ values that produce the target pyloric rhythm at each temperature. As in *Caplan et al., 2014* and *O'Leary and Marder, 2016* we searched for values of $Q_{10_i}$ between 1 and 2 for conductances and between 1 and 4 for activation/inactivation time scales. We built a new objective function by evaluating the previous objective function (*Equation 6*) over a set of control temperatures $T_i$ taken at 15℃, 20℃, 25℃ and 35℃,

$$E_C(\mathbf{Q_{10}}) = E(\mathbf{G}, T = 15°) + E(\mathbf{G}, T = 20°) + E(\mathbf{G}, T = 25°) + E_{crash}(\mathbf{G}, T = 35°). \qquad (7)$$

Here, $E(\mathbf{G}, T_i)$ is the score of the solutions of the set of conductances $\mathbf{G}$ at temperature $T_i$ and $E_{crash}$ is the total number of spikes in all three cells. This last term is introduced to enforce that when the temperature is close to 35℃ the network stops working as in experiments. Evaluation of the objective functions (*Equation 6*) and (*Equation 7*) requires that the model is simulated for a number of seconds and this is the part of the procedure that requires most computing power. Longer simulations will provide better estimations for the burst frequency and duty cycle of the solutions, but it will linearly increase the time it takes to evaluate them. If the simulations are shorter, evaluations of the objective function are faster but its minimization may be more difficult due to transient behaviors, and its minima may not correspond to stable pyloric rhythms.

Because we only target the activity at three temperature values, it is not guaranteed that the models will behave properly for temperatures in between. We found that if the system meets the target activity at the control temperatures, in the vast majority of the cases, the models also displayed the correct activity for temperatures in between, but there were exceptions (2 in 40 cases). This means that even if a solution achieves low scores, we are still required to screen the temperature values in between to make sure it is indeed temperature compensated over the full range. Increasing the number of control temperatures makes the estimation procedure slower and we met a good balance with 4 control temperatures. All models were subject to the same temperature perturbations and temperature compensation was confirmed in all cases by recording spikes and phases of bursting (*Figures 1* and *3*), membrane potential (*Figure 2*), and currents' dynamics (*Figures 7, 8* and *9*).

## Membrane potential and current shares distributions

The membrane potential distributions were computed for 101 values of temperature between 10℃ and 25℃. For each temperature, the models were simulated from identical initial conditions for 30 s with high numerical resolution ($dt = 0.001$). The distributions were computed using the last 10 s of each simulation. The number of samples of the distributions at each temperature is $5 \times 10^5$.

## Classification scheme

We classified the activity of each cell into several categories by inspecting their spiking patterns and features of their waveforms. For this we first compute the spike times and their ISI distributions. If the coefficient of variation $CV_{ISI} < 0.1$, we declare the activity as spiking. We then measure the lag $\delta$ between the spike onset and the crossing times with a slow wave threshold at $-50\,mV$. If this $\delta < 20\,msec$ we label the trace as *tonic spiking* and else, we label it as *single-spike bursting*. If $CV_{ISI} > 0.1$ then we ask if the cell is bursting. For this we group spikes into bursts using a temporal threshold of 200 ms and compute the distributions of duty cycle $dc$ and instantaneous burst frequency $f_b$. If $CV_{dc} < 1$, $CV_{f_b} < 0.1$ and $mean(dc) > 0$ we label the trace as *regular bursting*. If $mean(dc) > 0$ and either $CV_{dc} > 1$ or $CV_{f_b} > 0.1$ we label the trace as *irregular bursting*. If $mean(dc) = 0$ we label the trace as *irregular spiker*. If none of these conditions is met we label the trace as *other* and finally, if the total number of ISI values is not greater than 1 we label the trace as *quiescent*. The network state was classified as *triphasic* or *not triphasic*. We declared the network (NET) activity as *triphasic* if the frequency of the cells was similar (within 5%) and they fired in the right order, and not triphasic elsewhere.

## Parameters used in this study

Each model was assigned a six-character name. Here, we list which model was used in each figure. Code to simulate the network and the parameters of each model are supplemental to this work.

*Figure 1* 4 × 6W66. *Figure 1—figure supplement 1*: from left to right, WWZ3CN, FGFKQS, 71G6LA. *Figure 1—figure supplement 2*: PLKTEM. *Figure 1—figure supplement 3*: all models. *Figure 2*: G7P2WE. *Figure 2—figure supplement 1*: from left to right, G7P2WE, 4 × 6W66, MAXTTP. *Figure 3*: 4 × 6W66, SJR46Y.*Figure 4*: WWZ3CN. *Figure 5*: FGFKQS.*Figure 6*: FGFKQS, 71G6LA. *Figure 6—figure supplement 1*: in lexicographical order, B7SDOL, J9U8SQ, 4R2719, R3VIEW, Q8FBN1, RMZ8KZ, EF84RN, SJR46Y. *Figure 7*: (top to bottom), J9U8SQ, G81PUL, G7EOZ8, SAXT8Y, KNMLBC. *Figure 8*: OSHNMD, H5AIZR. *Figure 9*: WWZ3CN, FGFKQS. *Figure 10*: 4 × 6W66.

## Acknowledgements

LMA acknowledges useful conversations with Hillary Rodgers.

## Additional information

### Funding

| Funder | Grant reference number | Author |
| --- | --- | --- |
| National Institutes of Health | T32 NS07292 | Leandro M Alonso |
| National Institutes of Health | MH046742 | Eve Marder |
| National Institutes of Health | R35 NS097343 | Eve Marder |
| Swartz Foundation | Swartz Foundation 2017 | Leandro M Alonso |

The funders had no role in study design, data collection and interpretation, or the decision to submit the work for publication.

### Author contributions

Leandro M Alonso, Conceptualization, Software, Formal analysis, Investigation, Visualization, Writing - original draft, Writing - review and editing; Eve Marder, Conceptualization, Resources, Supervision, Funding acquisition, Writing - original draft, Writing - review and editing

### Author ORCIDs

Leandro M Alonso  https://orcid.org/0000-0001-8211-2855
Eve Marder  http://orcid.org/0000-0001-9632-5448

### Decision letter and Author response

Decision letter https://doi.org/10.7554/eLife.55470.sa1
Author response https://doi.org/10.7554/eLife.55470.sa2

## Additional files

### Supplementary files

- Transparent reporting form

### Data availability

Code and parameters to reproduce the results in this work was uploaded to Dryad, http://doi.org/10.5061/dryad.m31b1h7.

The following dataset was generated:

| Author(s) | Year | Dataset title | Dataset URL | Database and Identifier |
| --- | --- | --- | --- | --- |
| Alonso LM, Marder E | 2020 | Data from: Temperature compensation in a small rhythmic circuit | http://doi.org/10.5061/dryad.m31b1h7 | Dryad Digital Repository, 10.5061/dryad.m31b1h7 |

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
