## [Decision Letter]

**Acceptance summary:**

Taking advantage of the pyloric network in crustaceans, this computational study beautifully shows how circuits and behaviour can adapt to the effect of temperature changes as a result of smooth adaptations in cellular and channel protein mechanisms, thus preserving the overall function of the circuit.

**Decision letter after peer review:**

[Editors’ note: the authors submitted for reconsideration following the decision after peer review. What follows is the decision letter after the first round of review.]

Thank you for submitting your work entitled "Temperature compensation in a small rhythmic circuit" as a Research Advance for consideration by *eLife*. Your article has been reviewed by three peer reviewers, one of whom is a member of our Board of Reviewing Editors, and the evaluation has been overseen by a Senior Editor. The reviewers have opted to remain anonymous.

Our decision has been reached after an extensive discussion between the reviewers. Based on these discussions and the individual reviews below, we regret to inform you that your work will not be considered further for publication in *eLife*. As you will see, this was not an easy decision, but one in which the reviewers and Reviewing Editor reached consensus.

Summary

We viewed this work as a thorough demonstration of the complex effects of temperature, but the results did not represent enough of a conceptual or mechanistic advance as presented. In essence, although all the authors found the work interesting and visually stunning, several concerns were raised, as detailed below in the reviewers' comments. Further, even though it was felt that the authors could potentially address some of the concerns raised (e.g., re Q_10_ variation assumptions etc.), all of the reviewers felt that something more besides a descriptive presentation was warranted to justify publication in *eLife*.

Reviewer #1:

Since the paper that this work is building on is a tools and methods paper, we do not think that it is necessarily appropriate to be considered as a research advance paper, because A) it does not appear to be building on any of the tools and modeling techniques established in their previous publication, and B) it is using those techniques to explore a research topic that was not explored in the original publication. In other words, it would seem to be more appropriate to considered novel research, and not a research advance on the previous tools and methods paper. In their cover letter, the authors also considered this as an option. Our review is considered in the context of a research article.

In this manuscript, the authors use specialized computational methodologies and plotting techniques to uncover how current contributions in a set of models of the pyloric network change with temperature while still robustly generating stereotypical electrophysiological outputs. In selecting the models using their landscape optimization + a screening process approach, they ensured that the models were first capable of robustly generating triphasic spiking in the right order across different temperatures before analyzing current mechanisms through which this output emerges. For these models, they highlight a gradual change in current contributions as temperature is altered, showing that these current changes are complex and do not follow the same trends across models. In essence, they are able to show the existence of multistabilty, hysteresis and in general the non-trivial responses to changes in temperature.

The Marder lab has produced several interesting experimental and modeling studies examining temperature perturbations in previous works and this computational study adds to it. While the modeling work is robustly performed, it was thought that certain aspects of the modeling could be better contextualized in their connections to experimental literature. For example, when looking at hysteresis or removal of certain conductances during temperature changes, do the models – which were not designed to replicate any experimental electrophysiology outputs in these contexts – exhibit changes that are in line with what is seen experimentally (i.e. have there been any comparable experiments performed)? Could the authors consider/describe and/or design experiments that could be performed in light of their findings? For example, the finding that h-currents are not sensitive (and CaT are sensitive) to temperature (Figure 14C). Or perhaps a take-home finding would be that some types of currents (and not others) are sensitive? Is this something that the authors think appropriate in their context?

Reviewer #2:

This study compares models that reproduce operation of the pyloric network across a range of temperatures based on different sets of parameter values. Rather than identifying models that reproduce network operation using different sets of conductance densities (G), they identify models with different sets of G values and Q_10_ values. They then use their currentscape visualization approach to ascertain that "temperature changes the relative contributions of the current to neuronal activity so that rhythmic activity smoothly slides through changes in mechanisms." Though interesting, there has been a lot of work in this area (i.e. temperature compensation in the pyloric network in particular, and more generally about degeneracy in the STG), which impacts the novelty of the current study. Moreover, I have significant reservations about how Q_10_ values are treated.

Main Concerns:

1) To my knowledge, the Q_10_ value for a given process (single channel conductance or gating) in a certain channel type tends to be quite consistent, which means it shouldn't differ much between two individuals of the same species. But if I understand the current study, the Q_10_ value for a given process is constant across the cells of a given network, but varies between networks. Is that correct? If so, is that justified? This comes up in the Discussion, where a handful of papers about RNA editing are cited. But in Garrett and Rosenthal, 2012, the difference is observed in different (arctic vs tropical) species of octopus. Reenan et al., 2015, address acute temperature effects, but don't report any physiological (Q_10_) changes. Is there any evidence for this sort of RNA editing in crabs/lobsters? Variation in Q_10_ values across individual is a critical yet dubious assumption. At the very least, the authors need to be transparent about this at the start of the paper.

2) On the flip side, conductance densities within a network are fixed (i.e. do not vary with temperature). Is that correct? I thought that compensation was mediated via homeostatic changes mediated by changes in ion channel expression (O'Leary and Marder, 2016).

3) There are 37 conductance densities in the model, plus 14 different Q_10_ values. Even before variations in Q_10_ values are considered, I suspect that there are many different G values that can produce models that behave appropriately across the full temperature range. Indeed, the solution space is high dimensional, which allows for a variety of solutions. It is not clear to me if the even higher dimensionality afforded by variations in Q_10_ is really necessary. For instance, if the analysis presented in Figure 4 of this paper (to models with different sets of Q_10_ and G values) was applied to models with the same Q_10_ values but different G values, would the results be fundamentally different? Wouldn't the currentscapes still look different because of differences in G values (especially if G values were subject to homeostatic regulation)? I suspect there would still be smooth transitions between different relative fractions of currents. Showing that this doesn't happen without variation in Q_10_ values is important if a conclusion of this paper is that Q_10_ variation is critical.

Reviewer #3:

This work uses simulations of a reduced computational model of the crustacean pyloric network to make the important points that (a) neuronal networks that generate multi-phase bursting rhythms may be able to do so robustly across temperature changes despite the fact that these changes have different effects on different currents, and (b) the mechanisms underlying rhythmogenesis, including relative importance of currents, may change with temperature. The work mostly uses methods first presented by the authors in a 2019 *eLife* paper but looks at new issues.

Given the past works by these authors and others, I do not find these points to be highly surprising, and the paper also has some flaws that I will discuss in more detail below. My general assessment is that the authors' vivid illustration of these points in a computational model is a worthy contribution to the literature, but not a major advance.

Substantive concerns:

1) Much of the paper is written as a show and tell rather than as a flowing narrative. The authors tend to start sentences and paragraphs with phrases like "Figure X shows…" instead of presenting ideas, with figures shown in support of these ideas. I think some significant rewriting is needed to rectify this issue.

2) One of the issues that the authors consider is the relative roles of different currents in rhythms associated with different models and temperatures. I am not sure about the logic here: if a current has a larger magnitude in one regime than another, can we necessarily conclude that it has become more important? It may be that current A has grown larger, but current B has grown larger still; or perhaps current C, which is redundant with A, has also grown larger, such that A is no longer essential for rhythm generation. The authors need to be more careful with how this issue is presented; ideally, they would demonstrate that the changes are not just there, but are also meaningful. In a similar vein, they are making a choice to emphasize features that change between regimes, but there are also features that change very little, and this invariance could be as important as – or more important than – the changes. Finally, the extent of the changes in currents is not quantified – it is just displayed visually across Figures 8-10 and Figure 11. I would like to see some quantification of the changes, and I am not convinced that Figures 8-10 are needed at all since, to me, Figure 11 (which is beautiful) is much more illustrative.

3) I think the Materials and methods section needs some improvements and additions. The authors should clarify how they set up and check their target features for a rhythm to be considered valid – are there specifications for each cell? for the network as a whole? what if activity is rhythmic but not every cell bursts on each cycle? how are temperatures between the 4 main values "surveyed", to ensure that rhythmicity persists? Also, the authors should provide experimental citations for target values used for duty cycles and conductance ranges. I would like some justification for the use of the non-standard reversal potential of -70 mV for glutamate and for why the leak conductance is varied over a much smaller range than other conductances. For those interested in reproducing the results, the authors should indicate what computing resources (desktop? cluster?.…) and coding environment were used and how long simulations took (roughly – are we talking hours, days, weeks?). Finally, how were weights chosen for Equation 6?

4) Although the figures overall are informative and aesthetically pleasing, some revisions and clarifications are needed:

a) Figure 1A: I don't understand the rationale for the phase labels used. Why does the "PD off" box enclose the time when the PD cell is active? Why is there overlap between "LP on" and "LP off"? And so on! Some explanations are needed. Also, I don't see 3 curves in the top panel of Figure 1C, and the "PD, LP, PY" labels are too far removed from the data in the middle panel.

b) Figure 2: Panel B shows 6 examples. It would be nice to have some sort of summary of the findings over many models, not just these 6.

c) Figure 3D: the caption erroneously suggests that the duty cycle is multi-valued in the pink box.

d) Figure 6 and associated text: The authors should indicate how they define "failure" and how they detect it computationally (see (3) above).

e) Figure 11 supplement is lovely but clarification is needed about what p(V) (probability of voltage) means here. I am not seeing any dramatic effects, which makes me wonder if there is any scientific reason to keep this figure.

f) This is a very long paper. I am not convinced that the PCA adds much, and it's especially not clear to me how to get solid information from Figure 12C.

[Editors’ note: further revisions were suggested prior to acceptance, as described below.]

Thank you for submitting your article "Temperature compensation in a small rhythmic circuit" for consideration by *eLife*. Your article has been reviewed by three peer reviewers, and the evaluation has been overseen by a Reviewing Editor and Gary Westbrook as the Senior Editor. The following individuals involved in review of your submission have agreed to reveal their identity: Farzan Nadim (Reviewer #2); Alexandre Guet-McCreight (Reviewer #3).

The reviewers have discussed the reviews with one another and the Reviewing Editor has drafted this decision to help you prepare a revised submission.

Summary:

This manuscript uses computational modeling to examine how neural circuits can produce consistent activity across a wide range of temperatures. The authors use a family of models of the crustacean pyloric network, based on prior work from the Marder lab and a recent paper of the same authors. The main finding is that similar activity in the network, at different temperatures, can arise from the dynamic contributions of different currents. Additionally, the study shows that different currents play a compensatory role for one another at different temperatures, and that strong perturbations to the system at different temperatures could result in very different outcomes. It is noteworthy that this detailed level of analysis of current contributions is not possible to do experimentally. Therefore, this work also offers a glimpse at the coordination of ion channel mechanisms that could be occurring beyond the scope of current-day experimental apparatuses.

Essential revisions:

The reviewers found significant improvement over the original version, and the work was found to be comprehensive and worthy. However, concerns still remain. Overall, it was felt that the work as presented was too much on the descriptive side, the current narrative did not reflect what the interesting and important points of the study are. Some insight into why/how the compensation works needs to be provided. Specifics regarding these points are detailed here, and the various minor points raised by the reviewers are given below.

1) The narrative of the results should be changed and arranged around the main points.

That is, the Results narrative should draw the reader's attention to the highlights of the findings and (when allowed) the details should be put in the supplemental figures and tables.

In particular the many details of all the models voltage ranges, duty cycles etc. could be put in figure supplements. For instance, Figure 8, is quite beautiful with its currentscapes, but it is unclear what the point of the figure is? What information is the reader is supposed to take from this? There's no useful information even in the legend. The one point they bring up in the Results is rather subtle (they could've at least put arrows or circles) and the main points they make in the Results are actually in the figure supplements.

2) A problematic point here is that temperature changes are modeled in such an unconstrained manner (with 31 free parameters) that it is difficult to know whether any of the trajectories along different temperatures correspond to real biological changes. Does an ionic current respond to the same temperature change with widely different Q_10_s across preparations? If every single parameter is variable, independent of the conditions imposed, then what is the point of even measuring it? Would solution trajectories across a range of temperatures show more consistent underpinnings if the Q_10_s were consistent for each ionic current? In short, I find the unconstrained fitting of the parameters until the right solution is found akin to overfitting a curve to a few data points, and then trying to glean information from that fit. Please speak to this issue in some capacity.

3) There are interesting and important points in the study: 1. compensation of currents for one another at different temperatures (which I presume is a form of homeostasis), 2. Smooth transitions of these currents for one another. 3. Different responses to the same perturbation at different temperatures. 4. Distinct ways that the system crashes at high temperatures and 5. The presence of temperature-dependent hysteresis. However, the narrative in the presentation of the results (especially in the first half) is mainly descriptive and detail oriented, rather than driven by questions.

4) If one point of the study is that getting similar solutions at different temperatures requires ionic currents to assume different levels of contribution, this could be set as a central question and shown more directly. What is learned from how the models "crash" at high temperature? There may be some lessons there, but this point is not properly analyzed or explored.

Similarly, with the hysteresis. What is the lesson and what does it have to do with temperature or temperature compensation, rather than simply that a set of parameters may result in multistability?

5) Subsection “Spiking patterns during temperature ramps” and Figure 3E: Although clearer in this updated manuscript, can the authors add something to the figure to make this more obvious (e.g. adding the number of spikes in each burst above each burst in the trace or adding an inset plot showing superimposed traces of a cycle with and a cycle without the extra spike). Without actually counting the number of spikes in each cycle it's not obvious to me just from looking at the trace.

6) Paragraph three subsection “Spiking patterns during temperature ramps”: If these spiking patterns can be compared to experimental recordings, why is it the case that they are not compared here? In fact, this paragraph appears to end abruptly without much dissemination of Figure 4. I'm also hesitant on calling the duty cycle "temperature invariant", since it seems clear from Figure 4 that duty cycle definitely does vary with temperature to some degree – how the authors seem to define temperature-invariant strikes me as a bit too qualitative.

7) Paragraph six of subsection “Dynamics of the currents at different temperatures”: There are a lot of qualitative observations made regarding changes in currents across temperature and models. I feel as though this is an easy trap to fall into when there are many qualitative observations that could be made – for the reader, I find it becomes hard to follow since I end up going through the figures and squinting to try and see exactly what point the authors are trying to make for each observation. The main point is stated at the end and is very simple and much less specific: "Together, these examples illustrate how a current can play different roles at different temperatures, and how diverse these mechanisms can be across individual solutions". To bring this point forward better, I suggest the authors cut down on specific qualitative observations that individually do not carry much weight and speak more broadly. For example, you could replace the third sentence with something like "the current contributions profiles in model 2 are different leading up to and following bursts". Though I highlight this paragraph, I also feel that the writing is like this at various other points in the results as well.

8) Final paragraph of subsection “Dynamics of the currents at different temperatures”: Is there a case where current contribution transitions across temperature are not smooth? And is there any reason to believe that they would not be smooth in the first place? If not, I would argue that this does not seem like a very surprising finding given that many e-phys features are preserved and change gradually with temperature themselves. Of course, it is still informative as to why the e-phys features change gradually – just not really surprising. Perhaps some of the points put forth in the discussion should be first mentioned at this stage in the results to help the reader better understand why this finding is both important and non-intuitive.

[Editors’ note: further revisions were suggested prior to acceptance, as described below.]

Thank you for submitting your article "Temperature compensation in a small rhythmic circuit" for consideration by *eLife*. Your revised article has been reviewed by three peer reviewers, and the evaluation has been overseen by a Reviewing Editor and Gary Westbrook as the Senior Editor The following individuals involved in review of your submission have agreed to reveal their identity: Farzan Nadim (Reviewer #2); Alexandre Guet-McCreight (Reviewer #3).

The reviewers have discussed the reviews with one another and the Reviewing Editor has drafted this decision to help you prepare a revised submission.

We would like to draw your attention to changes in our revision policy that we have made in response to COVID-19 (https://elifesciences.org/articles/57162). Specifically, we are asking editors to accept without delay manuscripts, like yours, that they judge can stand as *eLife* papers without additional data, even if they feel that they would make the manuscript stronger. Thus, the revisions requested below only address clarity and presentation.

Summary:

This manuscript uses computational modeling to examine how neural circuits can produce consistent activity across a wide range of temperatures. The authors use a family of models of the crustacean pyloric network, based on prior work from the Marder lab and a recent paper of the same authors. The main finding is that similar activity in the network, at different temperatures, can arise from the dynamic contributions of different currents. Additionally, the study shows that different currents play a compensatory role for one another at different temperatures, and that strong perturbations to the system at different temperatures could result in very different outcomes.

Essential revisions:

All the reviewers found that the work was improved with better flow and was less descriptive. There are two aspects that the reviewers think should be addressed

1) Although the hysteresis result is very interesting, the motivation for looking at it in the context of temperature sensitivity/robustness still isn't provided, nor is the biological link. This aspect of the paper would benefit from providing more biological context about the significance of multistability with respect to temperature sensitivity.

2) Clarification of the authors' treatment of Q_10_s with a reasonable justification in the discussion, rather than hand waving. Specifically, although it is safe to assume that not all of the Q_10_ values are known and the authors set reasonable bounds when fitting the Q_10_ values, the concern is not that the Q_10_ for membrane current X sits somewhere between 1 and 4, but that the Q_10_ for membrane current X differs between models A, B and C when, in fact, the molecular identify of current X is the same in all of those models, because a Q_10_ value is not something that is regulated. Thus, if the authors took the set of Q_10_ values from model A or B or C, and then, using that set of fixed Q_10_ values, re-ran their genetic algorithm to determine new sets of conductance combinations that produce an acceptable triphasic rhythm, one might expect that successful conductance combinations will be far less diverse than when Q_10_ values are allowed to differ (as shown in Figure 1—figure supplement 1B). Without knowing the real Q_10_ values, we still won't know what the real conductance combinations could be, but we'd have a clearer picture of how diverse those combinations could be under biologically realistic conditions.

Although we encourage such additional simulations, it is not a requirement. However, an expanded discussion of how Q_10_'s are treated is warranted.

---

## [Author Response]

[Editors’ note: the authors resubmitted a revised version of the paper for consideration. What follows is the authors’ response to the first round of review.]

Reviewer #1:[…] The Marder lab has produced several interesting experimental and modeling studies examining temperature perturbations in previous works and this computational study adds to it. While the modeling work is robustly performed, it was thought that certain aspects of the modeling could be better contextualized in their connections to experimental literature. For example, when looking at hysteresis or removal of certain conductances during temperature changes, do the models – which were not designed to replicate any experimental electrophysiology outputs in these contexts – exhibit changes that are in line with what is seen experimentally (i.e. have there been any comparable experiments performed)? Could the authors consider/describe and/or design experiments that could be performed in light of their findings? For example, the finding that h-currents are not sensitive (and CaT are sensitive) to temperature (Figure 14C). Or perhaps a take-home finding would be that some types of currents (and not others) are sensitive? Is this something that the authors think appropriate in their context?

H-currents are sensitive too. Old Figure 14C in the original manuscript (now Figure 11) shows that removing this current results in similar activities at both 10 °C and 25 °C. Most models become quiescent upon removal of the H-current, and thus, removing it results in qualitatively similar activity.

Although we do observe some behaviors in common across models, the cohort is too small to make statements about which perturbation is most likely to result in a given behavior.

For the past year we have been trying to do experiments on hysteresis to temperature changes. These experiments are quite challenging because, as expected, the hysteresis depends sensitively on protocol, and of course, experimental preparations are never stationary. We are still struggling with these data, but have benefited by the guidance of these theoretical studies.

Reviewer #2:This study compares models that reproduce operation of the pyloric network across a range of temperatures based on different sets of parameter values. Rather than identifying models that reproduce network operation using different sets of conductance densities (G), they identify models with different sets of G values and Q_10_ values. They then use their currentscape visualization approach to ascertain that "temperature changes the relative contributions of the current to neuronal activity so that rhythmic activity smoothly slides through changes in mechanisms." Though interesting, there has been a lot of work in this area (i.e. temperature compensation in the pyloric network in particular, and more generally about degeneracy in the STG), which impacts the novelty of the current study. Moreover, I have significant reservations about how Q_10_ values are treated.

Despite extensive experimental work this is the first theoretical paper to address the potential mechanisms of phase compensation in response to temperature in the network. This is a fascinating issue and is present biologically, but non-trivial to capture in models.

Main Concerns:1) To my knowledge, the Q_10_ value for a given process (single channel conductance or gating) in a certain channel type tends to be quite consistent, which means it shouldn't differ much between two individuals of the same species. But if I understand the current study, the Q_10_ value for a given process is constant across the cells of a given network, but varies between networks. Is that correct? If so, is that justified? This comes up in the Discussion, where a handful of papers about RNA editing are cited. But in Garrett and Rosenthal, 2012, the difference is observed in different (arctic vs tropical) species of octopus. Reenan et al., 2015, address acute temperature effects, but don't report any physiological (Q_10_) changes. Is there any evidence for this sort of RNA editing in crabs/lobsters? Variation in Q_10_ values across individual is a critical yet dubious assumption. At the very least, the authors need to be transparent about this at the start of the paper.

We are implicitly assuming that the Q_10_ (which is a function of the protein structure) could vary across individuals if they show different isoforms of a channel protein, but within an individual it might be constant. That said, we do not assert that Q_10_ values are different across individuals, and we don’t know the experimental Q_10_ values of all the processes in our models. We show the existence of multiple mechanisms by which this can happen by finding Q_10_ values that permit the activity.

2) On the flip side, conductance densities within a network are fixed (i.e. do not vary with temperature). Is that correct? I thought that compensation was mediated via homeostatic changes mediated by changes in ion channel expression (O'Leary and Marder, 2016).

This is not completely correct. Conductance and the time scales of the processes both change acutely with temperature. This is our starting assumption. We show that there are many ways in which these quantities can change (Q_10_) without affecting the activity.

Homeostatic changes take place over a much longer time-scale. This work is about acute temperature perturbations that produce an instantaneous effect on the ion channels. This is now explicitly dealt with in the new Discussion.

3) There are 37 conductance densities in the model, plus 14 different Q_10_ values. Even before variations in Q_10_ values are considered, I suspect that there are many different G values that can produce models that behave appropriately across the full temperature range. Indeed, the solution space is high dimensional, which allows for a variety of solutions. It is not clear to me if the even higher dimensionality afforded by variations in Q_10_ is really necessary. For instance, if the analysis presented in Figure 4 of this paper (to models with different sets of Q_10_ and G values) was applied to models with the same Q_10_ values but different G values, would the results be fundamentally different? Wouldn't the currentscapes still look different because of differences in G values (especially if G values were subject to homeostatic regulation)? I suspect there would still be smooth transitions between different relative fractions of currents. Showing that this doesn't happen without variation in Q_10_ values is important if a conclusion of this paper is that Q_10_ variation is critical.

As we described above and in the new Discussion, we are not dealing with homeostatic compensation in this manuscript.

Reviewer #3:This work uses simulations of a reduced computational model of the crustacean pyloric network to make the important points that (a) neuronal networks that generate multi-phase bursting rhythms may be able to do so robustly across temperature changes despite the fact that these changes have different effects on different currents, and (b) the mechanisms underlying rhythmogenesis, including relative importance of currents, may change with temperature. The work mostly uses methods first presented by the authors in a 2019 eLife paper but looks at new issues.Given the past works by these authors and others, I do not find these points to be highly surprising, and the paper also has some flaws that I will discuss in more detail below. My general assessment is that the authors' vivid illustration of these points in a computational model is a worthy contribution to the literature, but not a major advance.

We have majorly rewritten the paper to make its new principles more evident.

Substantive concerns:1) Much of the paper is written as a show and tell rather than as a flowing narrative. The authors tend to start sentences and paragraphs with phrases like "Figure X shows…" instead of presenting ideas, with figures shown in support of these ideas. I think some significant rewriting is needed to rectify this issue.

Significant rewriting took place with special emphasis on the reviewer’s suggestion. Hopefully it reads much better now, thank you.

2) One of the issues that the authors consider is the relative roles of different currents in rhythms associated with different models and temperatures. I am not sure about the logic here: if a current has a larger magnitude in one regime than another, can we necessarily conclude that it has become more important? It may be that current A has grown larger, but current B has grown larger still; or perhaps current C, which is redundant with A, has also grown larger, such that A is no longer essential for rhythm generation. The authors need to be more careful with how this issue is presented; ideally, they would demonstrate that the changes are not just there, but are also meaningful. In a similar vein, they are making a choice to emphasize features that CHANGE between regimes, but there are also features that change very little, and this invariance could be as important as – or more important than – the changes. Finally, the extent of the changes in currents is not quantified – it is just displayed visually across Figures 8-10 and Figure 11. I would like to see some quantification of the changes, and I am not convinced that Figures 8-10 are needed at all since, to me, Figure 11 (which is beautiful) is much more illustrative.

We do not conclude that larger magnitude implies more importance. Moreover, we focus on the changes in percent contribution -not absolute value- to compare one current against the other and highlight changes.

It is true that even if a current contributes with only 1% of the total it could still be more important to the activity than another current that contributes more, and this is regardless of temperature. We do not attempt to establish a link between the magnitude of a currently and its importance by any definition. We refer as a current as “important" in the final paragraph of the Results where we explicitly show that deletion of a specific current is critical at one temperature but not the other.

The first half of the paper is about things that *stay the same* in the system, like duty cycle and phases of bursting. The second half is about *features that change*, like current contributions and so on. To demonstrate that the changes are not just there, but are also meaningful, we perform channel deletions and find that the responses are qualitatively different at different temperatures. If temperature scaled the currents by similar ratios then the activity should break down in a qualitatively similar way (only faster) at all temperatures. We find that in general this is not the case, and instead different current contributions result in different stability properties.

Old Figures 9 and 10 from the previous manuscript are now supplemental to Figure 8 of. The current version and Figure 11 (now Figure 9) was simplified to show only one model.

3) I think the Materials and methods section needs some improvements and additions. The authors should clarify how they set up and check their target features for a rhythm to be considered valid – are there specifications for each cell? for the network as a whole? what if activity is rhythmic but not every cell bursts on each cycle? how are temperatures between the 4 main values "surveyed", to ensure that rhythmicity persists? Also, the authors should provide experimental citations for target values used for duty cycles and conductance ranges. I would like some justification for the use of the non-standard reversal potential of -70 mV for glutamate and for why the leak conductance is varied over a much smaller range than other conductances. For those interested in reproducing the results, the authors should indicate what computing resources (desktop? cluster?.…) and coding environment were used and how long simulations took (roughly – are we talking hours, days, weeks?). Finally, how were weights chosen for Equation 6?

In subsection “Finding parameters” we state the target values for each cell and the network. We missed stating that the target frequency is 1 Hz. This is now fixed.

If the activity is rhythmic but one cell bursts every cycle or so, then the standard deviation of the duty cycle distribution is large and the solution is discarded. Minimization of Equation 6 produces sets of conductances for which the activity is rhythmic and all cells burst periodically as shown across this work.

The models are surveyed for temperatures in between the control temps. (10,20,25) in most figures. We performed the same analysis as in Figure 1 and Figure 1—figure supplement 1 for all models. We computed at the membrane potential distributions as in Figure 3 for all models taking 101 values between 10 °C and 25 °C. We subjected all models to continuous temperature ramps (see Figures 4 and 5) of different durations (see up ramps in Figure 5). We computed the distribution of currents over 101 values of temp. between 10 °C and 25 °C (current distributions not included in current version)

We included a citation to experimental values for the targets (Tang et al., 2010).

Glutamate is an inhibitory transmitter in the STG, and the glutamate synapses have reversal potentials of -65 mV to -70 mV (Marder and Eisen, 1984, Prinz et al., 2004). This is standard in this preparation.

We discretize our search space by taking 1000 equally spaced values between the bounds for each conductance. Thus, a larger domain is sampled with less resolution. The search space for the leak conductance is smaller because this conductance is typically much smaller than others.

We chose a search space that is large enough to contain variable solutions and that also contains some knowledge of where good solutions may be found to speed up our optimizations.

We used a commercially available desktop computer with 32-cores. Finding a control network from random initial parameters takes several hours. Finding Q_10_ values for a network takes another few hours.

Weights for Equation 6 were chosen by trial and error and were kept fixed throughout the work. Notice that duty cycle is on the order of 10^-1^, and this is squared so 10^-2^. Factor β=1000 makes it comparable to the other quantities, and maybe weighs burst duration more than other features.

Most of this information was included in the Materials and methods section of the original manuscript and is present here. Further information has been added in the current manuscript.

4) Although the figures overall are informative and aesthetically pleasing, some revisions and clarifications are needed:a) Figure 1A: I don't understand the rationale for the phase labels used. Why does the "PD off" box enclose the time when the PD cell is active? Why is there overlap between "LP on" and "LP off"? And so on! Some explanations are needed. Also, I don't see 3 curves in the top panel of Figure 1C, and the "PD, LP, PY" labels are too far removed from the data in the middle panel.

We added additional labels to make the figure clearer.

The reason there are not three curves in C-top is that the bursting frequency of the three cells is almost identical (as desired) and the dots overlap. This was clarified in the figure caption.

b) Figure 2: Panel B shows 6 examples. It would be nice to have some sort of summary of the findings over many models, not just these 6.

The figure was simplified to show 3 models where changes in overall amplitude are substantially different across them.

c) Figure 3D: the caption erroneously suggests that the duty cycle is multi-valued in the pink box.

Caption was rewritten. This figure was modified.

d) Figure 6 and associated text: The authors should indicate how they define "failure" and how they detect it computationally (see (3) above).

Failure is loss of triphasic rhythmic activity. We detected it by surveying the traces at different temperatures higher than 25 °C.

e) Figure 11 supplement is lovely but clarification is needed about what p(V) (probability of voltage) means here. I am not seeing any dramatic effects, which makes me wonder if there is any scientific reason to keep this figure.

Probability of V means the probability that you will measure V if you observe the system at a random instant. (blue is low probability, for example during spikes. It is more likely one will observe the system in between bursts than during a spike).

Changes across temperature (x-axis) are clear (panels B and C).

We concluded that this figure is too complicated and obscures the main points of the manuscript, and removed it for these reasons.

f) This is a very long paper. I am not convinced that the PCA adds much, and it's especially not clear to me how to get solid information from Figure 12C.

We agree with the reviewer. We removed several figures and the text that goes along with them. Figure 12 and the corresponding subsection were removed. We included a panel in Figure 8 showing how the contributions change across models.

[Editors’ note: what follows is the authors’ response to the second round of review.]

Essential revisions:The reviewers found significant improvement over the original version, and the work was found to be comprehensive and worthy. However, concerns still remain. Overall, it was felt that the work as presented was too much on the descriptive side, the current narrative did not reflect what the interesting and important points of the study are. Some insight into why/how the compensation works needs to be provided. Specifics regarding these points are detailed here, and the various minor points raised by the reviewers are given below.1) The narrative of the results should be changed and arranged around the main points.That is, the Results narrative should draw the reader's attention to the highlights of the findings and (when allowed) the details should be put in the supplemental figures and tables.In particular the many details of all the models voltage ranges, duty cycles etc. could be put in figure supplements. For instance, Figure 8, is quite beautiful with its currentscapes, but it is unclear what the point of the figure is? What information is the reader is supposed to take from this? There's no useful information even in the legend. The one point they bring up in the Results is rather subtle (they could've at least put arrows or circles) and the main points they make in the Results are actually in the figure supplements.

The Results section was heavily modified and rearranged. Much material was moved to the supplemental figures. The narrative was changed and several details are omitted.

2) A problematic point here is that temperature changes are modeled in such an unconstrained manner (with 31 free parameters) that it is difficult to know whether any of the trajectories along different temperatures correspond to real biological changes. Does an ionic current respond to the same temperature change with widely different Q_10_s across preparations? If every single parameter is variable, independent of the conditions imposed, then what is the point of even measuring it? Would solution trajectories across a range of temperatures show more consistent underpinnings if the Q_10_s were consistent for each ionic current? In short, I find the unconstrained fitting of the parameters until the right solution is found akin to overfitting a curve to a few data points, and then trying to glean information from that fit. Please speak to this issue in some capacity.

Our study is motivated by the experimental observation that neuronal circuits can remain performing their functions over a range of temperatures, at the same time that each of their molecular processes are affected differentially by temperature. Here we employ computational models to ask qualitatively how this can potentially take place in a network of neurons with multiple different types of ionic channels. For this we assumed a traditional Hodgkin and Huxley approach and asked if there are temperature robust solutions under the assumption that the temperature sensitivities Q_10_ of the different processes are different, and that the Q_10_ values are within experimental ranges (1 to 4 for timescales, 1 to 2 for conductances). We then employed the models to obtain qualitative insight into how compensation works and to formulate testable hypothesis. We are not aiming at extracting quantitative information about the pyloric network. Rather, we use the pyloric network to ask in general how this is possible.

3) There are interesting and important points in the study: 1. compensation of currents for one another at different temperatures (which I presume is a form of homeostasis), 2. Smooth transitions of these currents for one another. 3. Different responses to the same perturbation at different temperatures. 4. Distinct ways that the system crashes at high temperatures and 5. The presence of temperature-dependent hysteresis. However, the narrative in the presentation of the results (especially in the first half) is mainly descriptive and detail oriented, rather than driven by questions.

Homeostasis as currently described in the models produced by this lab (Liu et. al, 1998, O’Leary et. al, 2014) corresponds to changes in maximal conductances over time. In this study we are assuming that maximal conductances are fixed, so there is no homeostatic mechanism of any kind.

We assume this confusion stems from our previous past use of the term

“compensation” which evokes active changes in conductance densities. We now replaced the word “compensation” in most places by “robustness”, to highlight that here we are looking at automatic effects of neurons and networks that arise from the direct actions of temperature on ion channel proteins.

4) If one point of the study is that getting similar solutions at different temperatures requires ionic currents to assume different levels of contribution, this could be set as a central question and shown more directly. What is learned from how the models "crash" at high temperature? There may be some lessons there, but this point is not properly analyzed or explored.Similarly, with the hysteresis. What is the lesson and what does it have to do with temperature or temperature compensation, rather than simply that a set of parameters may result in multistability?

The Results section was modified to address this point. The “crashes” are now presented after we show currents. Hysteresis is also presented in that context.

5) Subsection “Spiking patterns during temperature ramps” and Figure 3E: Although clearer in this updated manuscript, can the authors add something to the figure to make this more obvious (e.g. adding the number of spikes in each burst above each burst in the trace or adding an inset plot showing superimposed traces of a cycle with and a cycle without the extra spike). Without actually counting the number of spikes in each cycle it's not obvious to me just from looking at the trace.

This figure was modified to highlight differences in the bursts. The figure was moved to the supplements (it is now Figure 1—figure supplement 2).

6) Paragraph three subsection “Spiking patterns during temperature ramps”: If these spiking patterns can be compared to experimental recordings, why is it the case that they are not compared here? In fact, this paragraph appears to end abruptly without much dissemination of Figure 4. I'm also hesitant on calling the duty cycle "temperature invariant", since it seems clear from Figure 4 that duty cycle definitely does vary with temperature to some degree – how the authors seem to define temperature-invariant strikes me as a bit too qualitative.

We decided to leave data out of this study as we are still working on these experiments.

The duty cycle changes with temperature but on average it stays close to a fixed value (red line in the plots). The y-axis scale in the plots was chosen to highlight these departures. We have a partial dataset which suggests the biological network also displays such departures and temperature dependence, but the lab was shut and will be many months until we can collect such data. In the context of *eLife* policy regarding COVID19 we want to go without data at this point and in the future we will come up with a full dataset.

Figure 4 has a thicker red line so small departures from the average value are now more evident. Also, this figure was moved to the supplements.

7) Paragraph six of subsection “Dynamics of the currents at different temperatures”: There are a lot of qualitative observations made regarding changes in currents across temperature and models. I feel as though this is an easy trap to fall into when there are many qualitative observations that could be made – for the reader, I find it becomes hard to follow since I end up going through the figures and squinting to try and see exactly what point the authors are trying to make for each observation. The main point is stated at the end and is very simple and much less specific: "Together, these examples illustrate how a current can play different roles at different temperatures, and how diverse these mechanisms can be across individual solutions". To bring this point forward better, I suggest the authors cut down on specific qualitative observations that individually do not carry much weight and speak more broadly. For example, you could replace the third sentence with something like "the current contributions profiles in model 2 are different leading up to and following bursts". Though I highlight this paragraph, I also feel that the writing is like this at various other points in the results as well.

We modified this section and removed much detail.

8) Final paragraph of subsection “Dynamics of the currents at different temperatures”: Is there a case where current contribution transitions across temperature are not smooth? And is there any reason to believe that they would not be smooth in the first place? If not, I would argue that this does not seem like a very surprising finding given that many e-phys features are preserved and change gradually with temperature themselves. Of course, it is still informative as to why the e-phys features change gradually – just not really surprising. Perhaps some of the points put forth in the discussion should be first mentioned at this stage in the results to help the reader better understand why this finding is both important and non-intuitive.

We did not see cases where the transitions are not smooth. However, we do not outrule that sharp transitions can take place in the working range, since maybe they are sharp but small. We do observe sharp transitions (possibly global bifurcations) when the models crash. We describe them as smooth because that’s how they are. It is unclear whether this is surprising or not.

[Editors’ note: further revisions were suggested prior to acceptance, as described below.]

Essential revisions:All the reviewers found that the work was improved with better flow and was less descriptive. There are two aspects that the reviewers think should be addressed1) Although the hysteresis result is very interesting, the motivation for looking at it in the context of temperature sensitivity/robustness still isn't provided, nor is the biological link. This aspect of the paper would benefit from providing more biological context about the significance of multistability with respect to temperature sensitivity.

Hysteresis mainly occurs when the models “crash" at high temperatures. The states such as those in Figure 7 in the previous version (now Figure 8) are the ones for which the system can display multistability and therefore hysteresis as temperature is ramped back and forth. Tang et al., 2010, did not report multistability in the working range (10 to 25 ^o^C), which is consistent with the models. That hysteresis occurs upon repeated “crashes” is a prediction of the models. This information was now added to the text, and is the subject of ongoing experiments in the lab. We are still struggling with the exact protocols to use to extract the most reliable insights from these experiments, as they are more technically difficult than might be expected.

2) Clarification of the authors' treatment of Q_10_s with a reasonable justification in the discussion, rather than hand waving. Specifically, although it is safe to assume that not all of the Q_10_ values are known and the authors set reasonable bounds when fitting the Q_10_ values, the concern is not that the Q_10_ for membrane current X sits somewhere between 1 and 4, but that the Q_10_ for membrane current X differs between models A, B and C when, in fact, the molecular identify of current X is the same in all of those models, because a Q_10_ value is not something that is regulated. Thus, if the authors took the set of Q_10_ values from model A or B or C, and then, using that set of fixed Q_10_ values, re-ran their genetic algorithm to determine new sets of conductance combinations that produce an acceptable triphasic rhythm, one might expect that successful conductance combinations will be far less diverse than when Q_10_ values are allowed to differ (as shown in Figure 1—figure supplement 1B). Without knowing the real Q_10_ values, we still won't know what the real conductance combinations could be, but we'd have a clearer picture of how diverse those combinations could be under biologically realistic conditions.Although we encourage such additional simulations, it is not a requirement. However, an expanded discussion of how Q_10_'s are treated is warranted.

Computational modeling will not answer what the “real” conductance and Q_10_ combinations are. Even the assumption that these quantities are stationary throughout the life of an animal (and/or the same across species) may not be warranted. For example, long-term acclimation over months resulting in RNA editing, could alter some Q_10_s. Moreover, it is possible that minor genetic variations in channel proteins exist in the population so that different animals in the population could show modestly different Q_10_s. Therefore, the point of this study is to ask generically how compensation occurs acutely assuming different temperature sensitivities for the processes, within the experimental range. We are not attempting to make quantitative observations about the pyloric network. We included a paragraph in the Discussion that makes this point more explicit.

We agree with the reviewer that if fixes one set of Q_10_s and asks for conductances that will produce temperature robust activity, this will be less diverse than if the Q_10_s are allowed to be different. We did perform some of these simulations, and it appears that the reviewer’s intuition is likely correct. However, quantifying this observation is tricky and beyond the scope of our work, and it overweighs the benefit of making such claim more precise.

In summary, we are not trying to find the one best solution. We want to have an overview of how these solutions can be, how diverse they can be, and what are the general features in common among them (such as differential stability, changes in contributions, etc.). Given all the compensated solutions we found share these qualitative features, we believe it is likely that the ‘true' biological mechanism also display these features. Our models provide testable predictions.